# Sensory cortex is optimized for prediction of future input

**Yosef Singer[1], Yayoi Teramoto[1], Ben DB Willmore[1], Jan WH Schnupp[2], Andrew J King[1], Nicol S Harper[1]***

[1]Department of Physiology, Anatomy and Genetics, University of Oxford, Oxford, United Kingdom; [2]Department of Biomedical Sciences, City University of Hong Kong, Kowloon Tong, Hong Kong

**Abstract** Neurons in sensory cortex are tuned to diverse features in natural scenes. But what determines which features neurons become selective to? Here we explore the idea that neuronal selectivity is optimized to represent features in the recent sensory past that best predict immediate future inputs. We tested this hypothesis using simple feedforward neural networks, which were trained to predict the next few moments of video or audio in clips of natural scenes. The networks developed receptive fields that closely matched those of real cortical neurons in different mammalian species, including the oriented spatial tuning of primary visual cortex, the frequency selectivity of primary auditory cortex and, most notably, their temporal tuning properties. Furthermore, the better a network predicted future inputs the more closely its receptive fields resembled those in the brain. This suggests that sensory processing is optimized to extract those features with the most capacity to predict future input.

DOI: https://doi.org/10.7554/eLife.31557.001

## Introduction

Sensory inputs guide actions, but such actions necessarily lag behind these inputs due to delays caused by sensory transduction, axonal conduction, synaptic transmission, and muscle activation. To strike a cricket ball, for example, one must estimate its future location, not where it is now (*Nijhawan, 1994*). Prediction has other fundamental theoretical advantages: a system that parsimoniously predicts future inputs from their past, and that generalizes well to new inputs, is likely to contain representations that reflect their underlying causes (*Bialek et al., 2001*). This is important because ultimately, we are interested in these causes (e.g. flying cricket balls), not the raw images or sound waves incident on the sensory receptors. Furthermore, much of sensory processing involves discarding irrelevant information, such as that which is not predictive of the future, to arrive at a representation of what is important in the environment for guiding action (*Bialek et al., 2001*).

Previous theoretical studies have suggested that many neural representations can be understood in terms of efficient coding of natural stimuli in a short time window at or just before the present (*Attneave, 1954*; *Barlow, 1959*; *Olshausen and Field, 1996*, *Olshausen and Field, 1997*). Such studies generally built a network model of the brain, which was trained to represent stimuli subject to some set of constraints. One pioneering such study trained a network to efficiently represent static natural images using a sparse, generative model (*Olshausen and Field, 1996*, *Olshausen and Field, 1997*). More recent studies have used related ideas to model the representation of moving (rather than static) images (*van Hateren and Ruderman, 1998a*; *Berkes and Wiskott, 2005*; *Berkes et al., 2009*) and other sensory stimuli (*Klein et al., 2003*; *Carlson et al., 2012*; *Zhao and Zhaoping, 2011*; *Kozlov and Gentner, 2016*; *Cusack and Carlyon, 2004*). In contrast, we built a network model that was optimized not for efficient representation of the recent past, but for efficient prediction of the immediate future of the stimulus, which we will refer to as the temporal

**\*For correspondence:**
nicol.harper@dpag.ox.ac.uk

**eLife digest** A large part of our brain is devoted to processing the sensory inputs that we receive from the world. This allows us to tell, for example, whether we are looking at a cat or a dog, and if we are hearing a bark or a meow. Neurons in the sensory cortex respond to these stimuli by generating spikes of activity. Within each sensory area, neurons respond best to stimuli with precise properties: those in the primary visual cortex prefer edge-like structures that move in a certain direction at a given speed, while neurons in the primary auditory cortex favour sounds that change in loudness over a particular range of frequencies.

Singer et al. sought to understand why neurons respond to the particular features of stimuli that they do. Why do visual neurons react more to moving edges than to, say, rotating hexagons? And why do auditory neurons respond more to certain changing sounds than to, say, constant tones? One leading idea is that the brain tries to use as few spikes as possible to represent real-world stimuli. Known as sparse coding, this principle can account for much of the behaviour of sensory neurons.

Another possibility is that sensory areas respond the way they do because it enables them to best predict future sensory input. To test this idea, Singer et al. used a computer to simulate a network of neurons and trained this network to predict the next few frames of video clips using the previous few frames. When the network had learned this task, Singer et al. examined the neurons' preferred stimuli. Like neurons in primary visual cortex, the simulated neurons typically responded most to edges that moved over time.

The same network was also trained in a similar way, but this time using sound. As for neurons in primary auditory cortex, the simulated neurons preferred sounds that changed in loudness at particular frequencies. Notably, for both vision and audition, the simulated neurons favoured recent inputs over those further into the past. In this way and others, they were more similar to real neurons than simulated neurons that used sparse coding.

Both artificial networks trained to foretell sensory input and the brain therefore favour the same types of stimuli: the ones that are good at helping to grasp future information. This suggests that the brain represents the sensory world so as to be able to best predict the future.

Knowing how the brain handles information from our senses may help to understand disorders associated with sensory processing, such as dyslexia and tinnitus. It may also inspire approaches for training machines to process sensory inputs, improving artificial intelligence.
DOI: https://doi.org/10.7554/eLife.31557.002

prediction model. The timescale of prediction considered for our model is in the range of tens to hundreds of milliseconds. Conduction delays to cortex and very fast motor responses are on this timescale (*Bixler et al., 1967*; *Yeomans and Frankland, 1995*; *Bizley et al., 2005*).

The idea that prediction is an important component of perception dates at least as far back as Helmholtz (*Helmholtz, 1962*; *Sutton and Barton, 1981*), although what is meant by prediction and the purpose it serves is quite varied between models incorporating it (*Chalk et al., 2018*; *Salisbury and Palmer, 2016*). With regards to perception and prediction, two contrasting but inter-related frameworks have been distinguished (*Chalk et al., 2018*; *Salisbury and Palmer, 2016*). In the 'predictive coding' framework (*Huang and Rao, 2011*; *Rao and Ballard, 1999*; *Friston, 2003*), prediction is used to remove statistical redundancy in order to provide an efficient representation of the entire stimulus. Some models of this type use prediction as a term for estimation of the current or a static input (such as images) from latent variables (*Rao and Ballard, 1999*), whereas other have also considered the temporal dimension of the input (*Rao and Ballard, 1997*; *Rao, 1999*; *Srinivasan et al., 1982*). Sparse coding models (*Olshausen and Field, 1996*, *Olshausen and Field, 1997*) can be related to this framework (*Huang and Rao, 2011*). In contrast, the 'predictive information' framework (*Bialek et al., 2001*; *Salisbury and Palmer, 2016*; *Palmer et al., 2015*; *Heeger, 2017*), which our approach relates to more closely, involves selective encoding of those features of the stimulus that predict future input. A related idea to predictive information is the encoding of slowly varying features (*Berkes and Wiskott, 2005*; *Creutzig and Sprekeler, 2008*; *Kayser et al., 2001*; *Hyvärinen et al., 2003*), which are one kind of predictive feature. Hence, the

predictive coding approach seeks to find a compressed representation of the entire input, whereas the predictive information approach selectivity encodes only predictive features (*Chalk et al., 2018*; *Salisbury and Palmer, 2016*). Our model relates to the predictive information approach in that it is optimized to predict the future from the past, but it has a combination of characteristics, such a non-linear encoder and sparse weight regularization, which have not previously been explored for such an approach.

To evaluate the representations produced by these normative theoretical models, they can be optimized for natural stimuli, and the tuning properties of their units compared to the receptive fields of real neurons. A useful and commonly used definition of a neuron's receptive field (RF) is the stimulus that maximally linearly drives the neuron (*Adelson and Bergen, 1985*; *Aertsen et al., 1981*; *Aertsen and Johannesma, 1981*; *Reid et al., 1987*; *deCharms et al., 1998*; *Harper et al., 2016*). In mammalian primary visual cortex (V1), neurons typically respond strongly to oriented edge-like structures moving over a particular retinal location (*Hubel and Wiesel, 1959*; *Jones and Palmer, 1987*; *DeAngelis et al., 1993*; *Ringach, 2002*). In mammalian primary auditory cortex (A1), most neurons respond strongly to changes in the amplitude of sounds within a certain frequency range (*deCharms et al., 1998*).

The temporal prediction model provides a principled approach to understanding the temporal aspects of RFs. Previous models, based on sparsity or slowness related principles, were successful in accounting for many spatial aspects of V1 RF structure (*Olshausen and Field, 1996*, *Olshausen and Field, 1997*; *van Hateren and Ruderman, 1998a*; *Berkes and Wiskott, 2005*; *Berkes et al., 2009*; *van Hateren and van der Schaaf, 1998b*), and had some success in accounting for spectral aspects of A1 RF structure (*Klein et al., 2003*; *Carlson et al., 2012*; *Zhao and Zhaoping, 2011*; *Cusack and Carlyon, 2004*). However, these models do not account well for the temporal structure of V1 or A1 RFs. Notably, for both vision (*Ringach, 2002*) and audition (*deCharms et al., 1998*), the envelopes of real neuronal RFs tend to be asymmetric in time, with greater sensitivity to very recent inputs compared to inputs further in the past. In contrast, the RFs predicted by previous models (*van Hateren and Ruderman, 1998a*; *Klein et al., 2003*; *Carlson et al., 2012*; *Kozlov and Gentner, 2016*; *Cusack and Carlyon, 2004*) typically show symmetrical temporal envelopes, with either approximately flat envelopes over time or a balanced falloff of the envelope over time either side of a peak. They also lack the greater sensitivity to very recent inputs.

Here we show using qualitative and quantitative comparisons that, for both V1 and A1 RFs, these shortcomings are largely overcome by the temporal prediction approach. This suggests that neural sensitivity at early levels of the cortical hierarchy may be organized to facilitate a rapid and efficient prediction of what the environment will look like in the next fraction of a second.

## Results

### The temporal prediction model

To determine what type of sensory RF structures would facilitate predictions of the imminent future, we built a feedforward network model with a single layer of nonlinear hidden units, mapping the inputs to the outputs through weighted connections (*Figure 1*). Each hidden unit's output results from a linear mapping (by input weights) from the past input, followed by a monotonic nonlinearity, much like the classic linear-nonlinear model of sensory neurons (*Klein et al., 2003*; *Carlson et al., 2012*; *Zhao and Zhaoping, 2011*). The model then generates a prediction of the future from a linear mapping (by output weights) from the hidden units' outputs. This is consistent with the observation that decoding from the neural response is often well approximated by a linear transformation (*Eliasmith and Anderson, 2003*).

We trained the temporal prediction model on extensive corpora, either of soundscapes or silent movies, modelling A1 (*Figure 1a*) or V1 (*Figure 1b*) neurons, respectively. In each case, the networks were trained by optimizing their synaptic weights to most accurately predict the immediate future of the stimulus from its very recent past. For vision, the inputs were patches of videos of animals moving in natural settings, and we trained the network to predict the pixel values for one movie frame (40 ms) into the future, based on the seven most recent frames (280 ms). For audition, we trained the network to predict the next three time steps (15 ms) of cochleagrams of natural sounds based

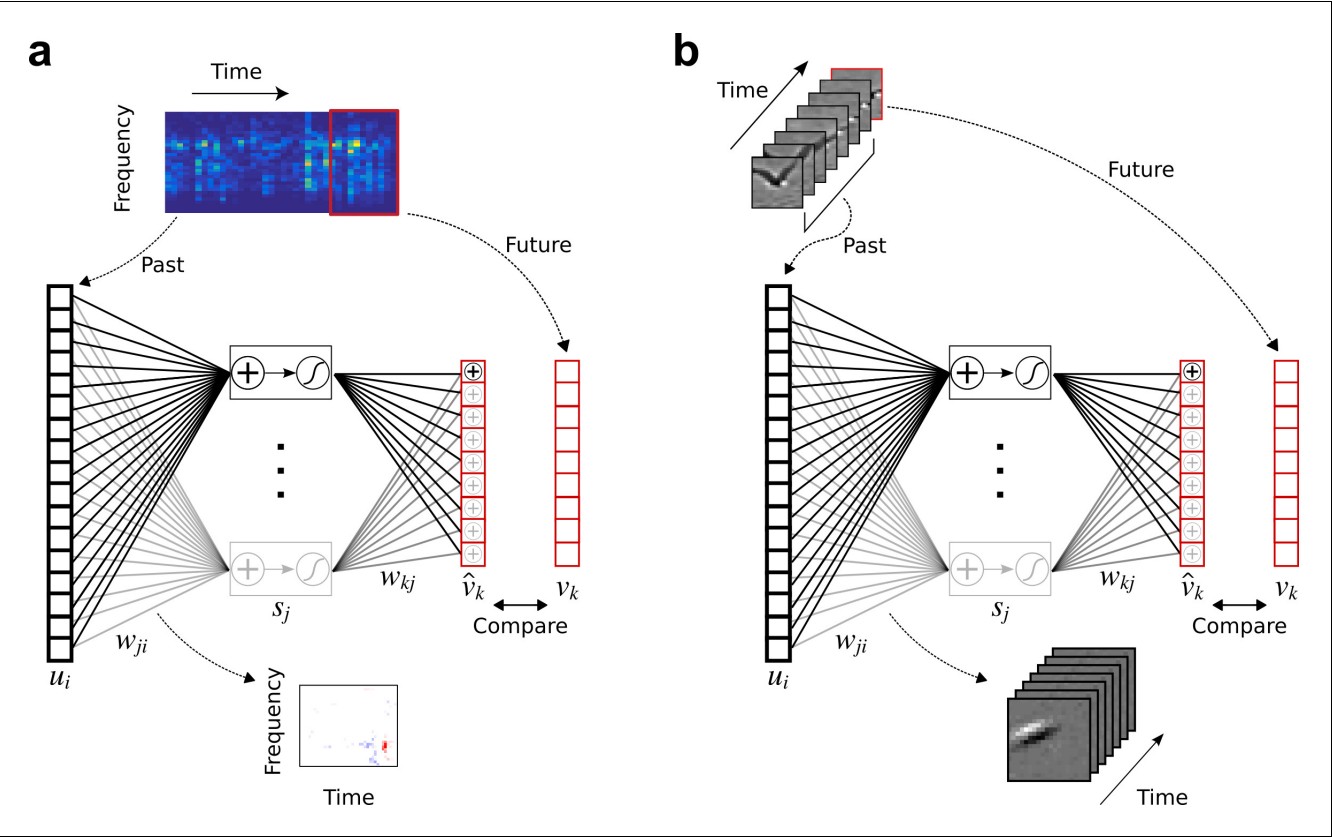

**Figure 1.** Temporal prediction model implemented using a feedforward artificial neural network, with the same architecture in both visual and auditory domains. (**a**), Network trained on cochleagram clips (spectral content over time) of natural sounds, aims to predict immediate future time steps of each clip from recent past time steps. (**b**), Network trained on movie clips of natural scenes, aims to predict immediate future frame of each clip from recent past frames. $u_i$, input – the past; $w_{ji}$, input weights; $s_j$, hidden unit output; $w_{kj}$, output weights; $\hat{v}_k$, output – the predicted future; $v_k$, target output – the true future. Hidden unit's RF is the $w_{ji}$ between the input and that unit $j$.

DOI: https://doi.org/10.7554/eLife.31557.003

on the 40 most recent time steps (200 ms). Cochleagrams resemble spectrograms but are adjusted to approximate the auditory nerve representation of sounds (see Materials and methods).

During training we used sparse, $L_1$ weight regularization (see *Equation 3* in Materials and methods) to constrain the network to predict future stimuli in a parsimonious fashion, forcing the network to use as few weights as possible while maintaining an accurate prediction. This constraint can be viewed as an assumption about the sparse nature of causal dependencies underlying the sensory input, or alternatively as analogous to the energy and space restrictions of neural connectivity. It also prevents our network model from overfitting to its inputs. Note that this sparsity constraint differs from that used in sparse coding models, in that it is applied to the weights rather than the activity of the units, being more like a constraint on the wiring between neurons than a constraint on their firing rates.

## Qualitative assessment of auditory receptive fields

To compare with the model, we recorded responses of 114 auditory neurons (including 76 single units) in A1 and the anterior auditory field (AAF) of 5 anesthetized ferrets (*Willmore et al., 2016*) and measured their spectrotemporal RFs (see Materials and methods). Ferrets are commonly used for auditory research, because they are readily trained in a range of sound detection, discrimination or localization tasks (*Nodal and King, 2014*), the frequency range of their hearing (approximately 40 Hz–40 kHz [*Kavanagh and Kelly, 1988*]) overlaps well with (and extends beyond) the human range, and most of their auditory cortex is not buried in a sulcus and hence easily accessible for electrophysiological or optical measurements.

The A1 RFs we recorded are diverse (*Figure 2a*); their frequency tuning can be narrowband or broadband, and sometimes showing flanking inhibition. Some may also be more complex in frequency tuning, lack clear order, or be selective for the direction of frequency modulation (*Carlin and Elhilali, 2013*).

In their temporal tuning, A1 RFs tend to weight recent inputs more heavily, with a temporally asymmetric power profile, involving excitation near the present followed by lagging inhibition of a longer duration (*deCharms et al., 1998*). The temporal prediction model RFs (*Figure 2b*) are similarly diverse, showing all of the RF types seen in vivo (including examples of localized, narrowband, broadband, complex, disordered and directional RFs) and are well matched in scale and form to those measured in A1. This includes having greater power (mean square) near the present, with brief excitation followed by longer lagging inhibition, producing an asymmetric power profile. This stands in contrast to previous attempts to model RFs based on efficient coding,sparsecoding and slow feature hypotheses, which either did not capture the diversity of RFs (*Zhao and Zhaoping, 2011*), or lacked temporal asymmetry, punctate structure, or appropriate time scale (*Klein et al., 2003*; *Carlson et al., 2012*; *Kozlov and Gentner, 2016*; *Cusack and Carlyon, 2004*; *Carlin and Elhilali, 2013*; *Brito and Gerstner, 2016*).

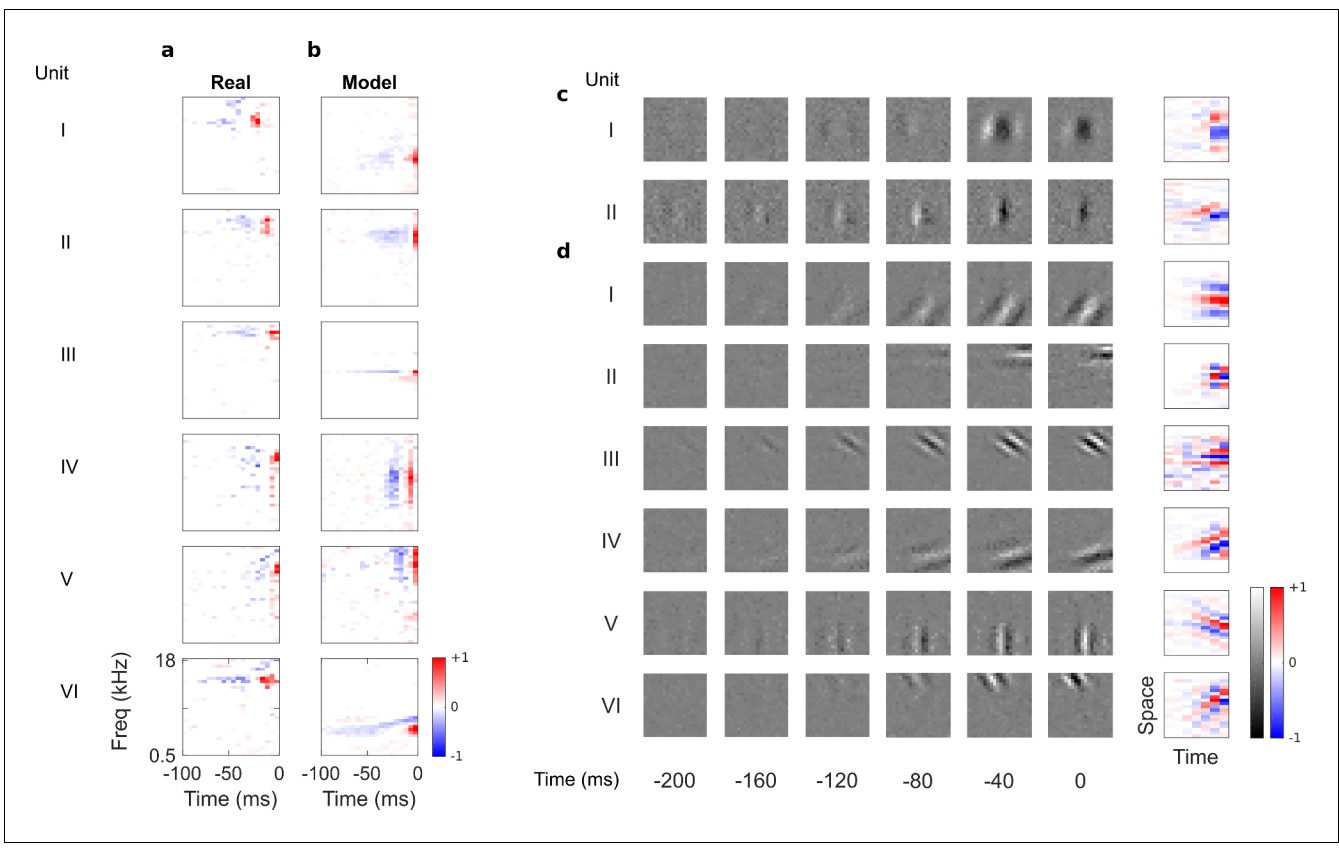

**Figure 2.** Auditory spectrotemporal and visual spatiotemporal RFs of real neurons and temporal prediction model units. (**a**), Example spectrotemporal RFs of real A1 neurons (*Willmore et al., 2016*). Red – excitation, blue – inhibition. Most recent two time steps (10 ms) were removed to account for conduction delay. (**b**), Example spectrotemporal RFs of model units when model is trained to predict the future of natural sound inputs. Note that the overall sign of a receptive field learned by the model is arbitrary. Hence, in all figures and analyses we multiplied each model receptive field by −1 where appropriate to obtain receptive fields which all have positive leading excitation (see Materials and methods). (**c**), Example spatiotemporal (I, space-time separable, and II, space-time inseparable) RFs of real V1 neurons (*Ohzawa et al., 1996*). Left, grayscale: 3D (space-space-time) spatiotemporal RFs showing the spatial RF at each of the most recent six time steps. Most recent time step (40 ms) was removed to account for conduction delay. White – excitation, black – inhibition. Right: corresponding 2D (space-time) spatiotemporal RFs obtained by summing along the unit's axis of orientation for each time step. Red – excitation, blue – inhibition. (**d**), Example 3D and corresponding 2D spatiotemporal (I-III, space-time separable, and IV-VI, space-time inseparable) RFs of model units when model is trained to predict the future of natural visual inputs.
DOI: https://doi.org/10.7554/eLife.31557.004

## Qualitative assessment of visual receptive fields

By eye, substantial similarities were also apparent when we compared the temporal prediction model's RFs trained using visual inputs (*Figure 1b*) with the 3D (space-space-time) and 2D (space-time) spatiotemporal RFs of real V1 simple cells, which were obtained from Ohzawa et al (*Ohzawa et al., 1996*). Simple cells (*Hubel and Wiesel, 1959*) have stereotyped RFs containing parallel, spatially localized excitatory and inhibitory regions, with each cell having a particular preferred orientation and spatial frequency (*Jones and Palmer, 1987*; *DeAngelis et al., 1993*; *Ringach, 2002*) (*Figure 2c*). These features are also clearly apparent in the model RFs (*Figure 2d*).

Unlike previous models (*van Hateren and Ruderman, 1998a*; *Hyvärinen et al., 2003*; *Olshausen, 2003*), the temporal prediction model captures the temporal asymmetry of real RFs. The RF power is highest near the present and decays into the past (*Figure 2d*), as observed in real neurons (*Ohzawa et al., 1996*) (*Figure 2c*). Furthermore, simple cell RFs have two types of spatiotemporal structure: space-time separable RFs (*Figure 2cI*), whose optimal stimulus resembles a flashing or slowly ramping grating, and space-time inseparable RFs, whose optimal stimulus is a drifting grating (*DeAngelis et al., 1993*) (*Figure 2cII*). Our model captures this diversity (*Figure 2dI–III* separable, *Figure 2dIV–VI* inseparable).

We also examined linear aspects of the tuning of the output units for the visual temporal prediction model using a response-weighted average to white noise input, and found punctate non-oriented RFs that decay into the past.

## Qualitative comparison to other models

For comparison, we trained a sparse coding model (*Olshausen and Field, 1996*, *Olshausen and Field, 1997*; *Carlson et al., 2012*) (https://github.com/zayd/sparsenet) using our dataset. We would expect such a model to perform less well in the temporal domain, because unlike the temporal prediction model, the direction of time is not explicitly accounted for. The sparse coding model was chosen because it has set the standard for normative models of visual RFs (*Olshausen and Field, 1996*, *Olshausen, 2003*; *Olshausen and Field, 1997*), and the same model has also been applied for auditory RFs (*Carlson et al., 2012*; *Brito and Gerstner, 2016*; *Młynarski and McDermott, 2017*; *Blättler et al., 2011*). Past studies (*Olshausen and Field, 1996*, *Olshausen and Field, 1997*; *Carlson et al., 2012*) have largely analysed the basis functions produced by the sparse coding model and compared their properties to neuronal RFs. To be consistent with these studies we have done the same, and to have a common term, refer to the basis functions as RFs (although strictly, they are projective fields). We can visually compare the large set of RFs recorded from A1 neurons (*Figure 3*) to the full set of RFs obtained from the temporal prediction model when trained on auditory inputs (*Figure 4*) and those of the sparse coding model (*Figure 5*) when trained on the same auditory inputs.

A range of RFs were produced by the sparse coding model, some of which show characteristics reminiscent of A1 RFs, particularly in the frequency domain. However, the temporal properties of A1 neurons are not well captured by these RFs. While some RFs display excitation followed by lagging inhibition, very few, if any, show distinct brief excitation followed by extended inhibition. Instead, RFs that show both excitation and inhibition tend to have a symmetric envelope and these features are randomly localized in time, and many RFs display temporally elongated structures that are not found in A1 neurons.

We also trained the sparse coding model on the dataset of visual inputs to serve as a control for the temporal prediction model trained on these same inputs. We compared the full population of spatial and 2D spatiotemporal visual RFs of the temporal prediction model (*Figure 4—figure supplements 2–3*) and the sparse coding model (*Figure 5—figure supplements 1–2*). As shown in previous studies (*Olshausen and Field, 1996*, *Olshausen and Field, 1997*; *van Hateren and Ruderman, 1998a*; *van Hateren and van der Schaaf, 1998b*), the sparse coding model produces RFs whose spatial structure resembles that of V1 simple cells (*Figure 5—figure supplements 1–2*), but does not capture the asymmetric nature of the temporal tuning of V1 neurons. Furthermore, while it does produce examples of both separable and inseparable spatiotemporal RFs, those that are separable tend to be completely stationary over time, resembling immobile rather than flashing gratings (*Figure 5—figure supplement 2*).

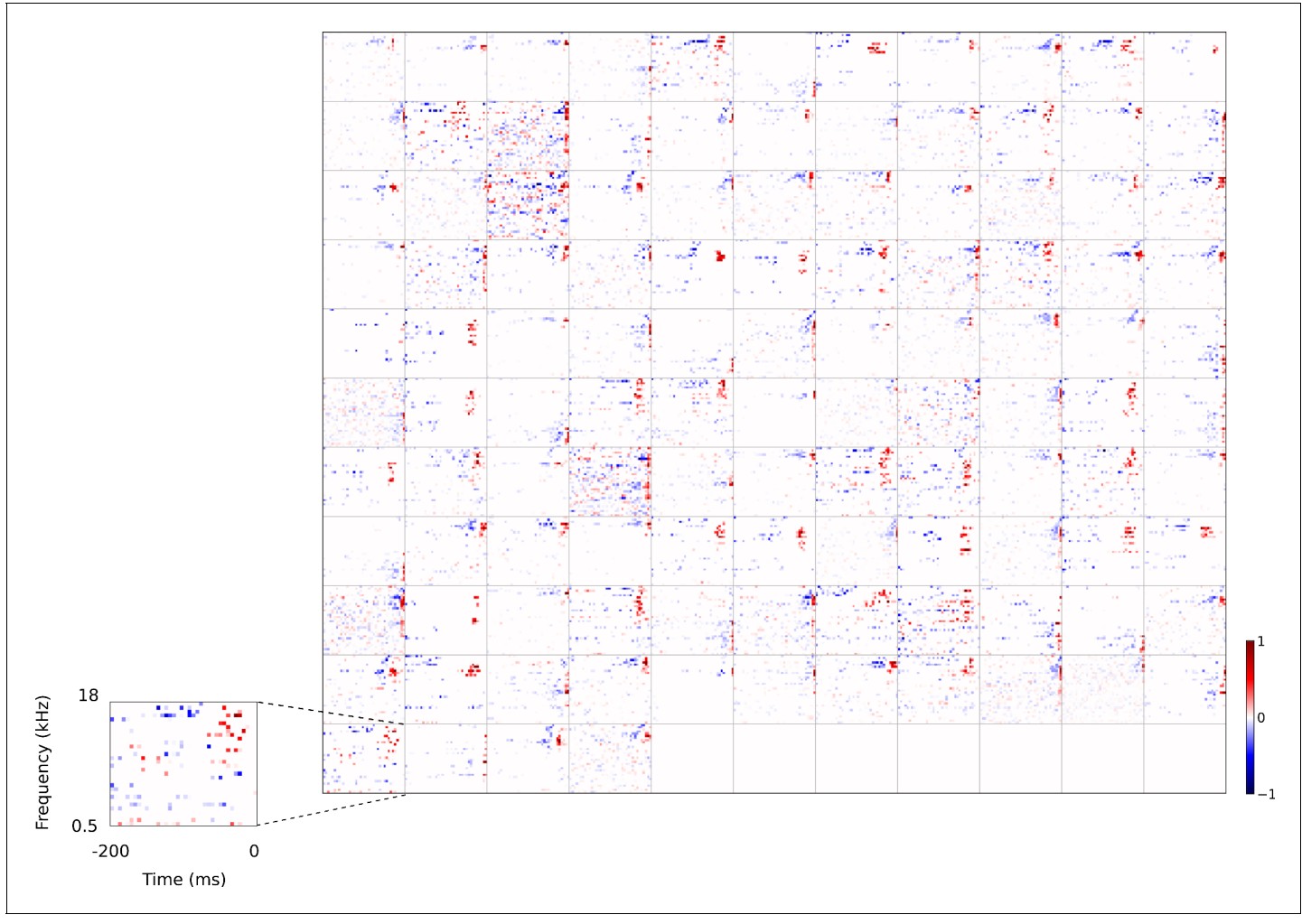

**Figure 3.** Full dataset of real auditory RFs. 114 neuronal RFs recorded from A1 and AAF of 5 ferrets. Red – excitation, blue - inhibition. Inset shows axes.

DOI: https://doi.org/10.7554/eLife.31557.005

## Quantitative analysis of auditory results

We compared the RFs generated by both models to the RFs of the population of real A1 neurons we recorded. We first compared the RFs in a non-parametric manner by measuring the Euclidean distances between the coefficient values of the RFs, and then used multi-dimensional scaling to embed these distances in a two-dimensional space (*Figure 6a*). The RFs of the sparse coding model span a much larger region than the real A1 and temporal prediction model RFs. Furthermore, the A1 and temporal prediction model RFs occupy a similar region of the space, indicating their greater similarity to each other relative to those of the sparse coding model. We then examined specific attributes of the RFs to determine points of similarity and difference between each of the models and the recorded data. We first considered the temporal properties of the RFs and found that for the data and the temporal prediction model, most of the power is contained in the most recent time-steps (*Figures 2a–b*, *3–4* and *6b*, and *Figure 4—figure supplement 1*). Given that the direction of time is not explicitly accounted for in the sparse coding model, as expected, it does not show this feature (*Figures 5* and *6b*). Next, we examined the tuning widths of the RFs in each population for both time and frequency, looking at excitation and inhibition separately. In the time domain, the real data tend to show leading excitation followed by lagging inhibition of longer duration (*Figures 2a*, *3* and *6c–e*). The temporal prediction model also shows many RFs with this temporal structure, with lagging inhibition of longer duration than the leading excitation (*Figures 2b*,

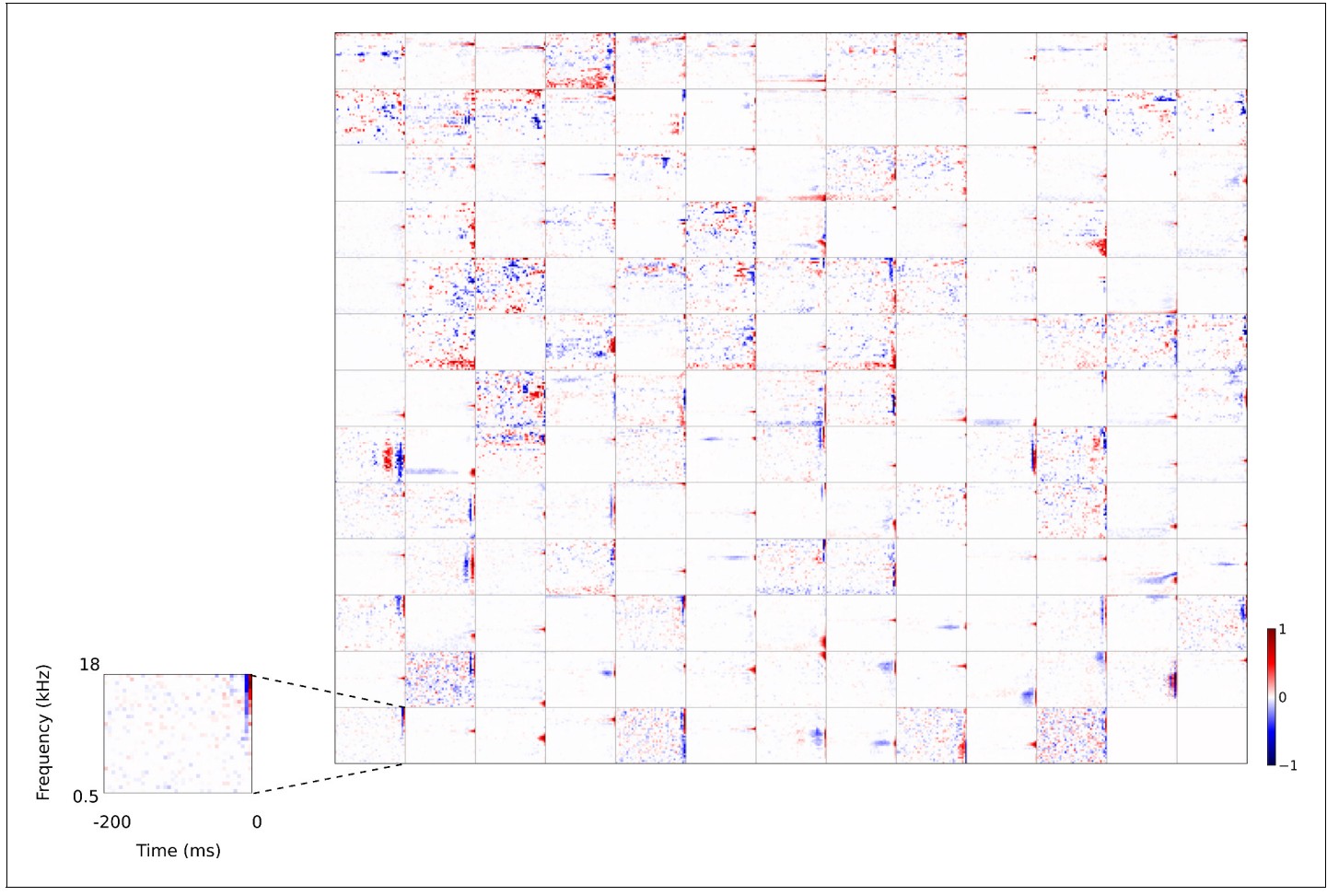

**Figure 4.** Full set of auditory RFs of the temporal prediction model units. Units were obtained by training the model with 1600 hidden units on auditory inputs. The hidden unit number and $L_1$ weight regularization strength ($10^{-3.5}$) was chosen because it results in the lowest MSE on the prediction task, as measured using a cross validation set. Many hidden units' weight matrices decayed to near zero during training (due to the $L_1$ regularization), leaving 167 active units. Inactive units were excluded from analysis and are not shown. Example units in *Figure 2* come from this set. Red – excitation, blue - inhibition. Inset shows axes. *Figure 4—figure supplement 1* shows the same RFs on a finer timescale. The full sets of visual spatial and corresponding spatiotemporal RFs for the temporal prediction model when it is trained on visual inputs are shown in *Figure 4—figure supplements 2–3*. *Figure 4— figure supplement 4* shows the auditory RFs of the temporal prediction model when a linear activation function instead of a sigmoid nonlinearity was used. *Figure 4—figure supplement 5–7* show the auditory spectrotemporal and visual spatial and 2D spatiotemporal RFs of the temporal prediction model when it was trained on inputs without added noise.

DOI: https://doi.org/10.7554/eLife.31557.006

The following figure supplements are available for figure 4:

**Figure supplement 1.** Full set of auditory RFs of the temporal prediction model units shown on a finer timescale.
DOI: https://doi.org/10.7554/eLife.31557.007

**Figure supplement 2.** Full set of visual spatial RFs of the temporal prediction model units.
DOI: https://doi.org/10.7554/eLife.31557.008

**Figure supplement 3.** Visual 2D (space-time) spatiotemporal RFs of temporal prediction model units.
DOI: https://doi.org/10.7554/eLife.31557.009

**Figure supplement 4.** Full set of auditory RFs of the temporal prediction model units using a linear activation function.
DOI: https://doi.org/10.7554/eLife.31557.010

**Figure supplement 5.** Full set of auditory RFs of the temporal prediction model units trained on auditory inputs without added noise.
DOI: https://doi.org/10.7554/eLife.31557.011

**Figure supplement 6.** Full set of visual spatial RFs of temporal prediction model units trained on visual inputs without added noise.
DOI: https://doi.org/10.7554/eLife.31557.012

**Figure supplement 7.** 2D (space-time) visual spatiotemporal RFs of temporal prediction model units trained on visual inputs without added noise.
DOI: https://doi.org/10.7554/eLife.31557.013

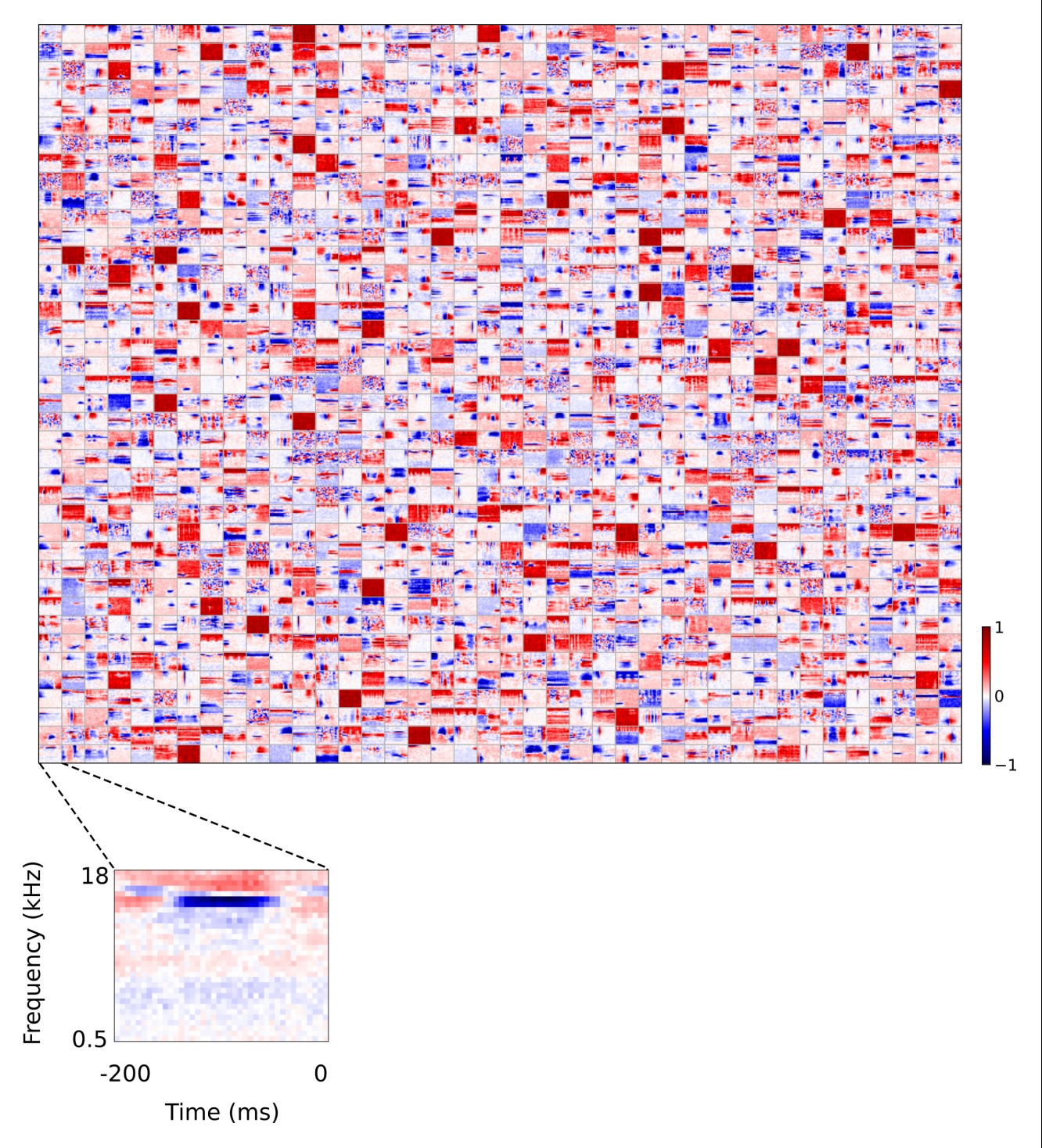

**Figure 5.** Full set of auditory 'RFs' (basis functions) of sparse coding model used as a control. Units were obtained by training the sparse coding model with 1600 units on the identical auditory inputs used to train the network shown in **Figure 4**. $L_1$ regularization of strength $10^{0.5}$ was applied to the units' activities. This network configuration was selected as it produced unit RFs that most closely resembled those recorded in A1, as determined using the KS measure of similarity **Figure 8—figure supplement 1** . Although the basis functions of the sparse coding model are not receptive fields, but projective fields, they tend to be similar in structure (**Olshausen and Field, 1996**, **Olshausen and Field, 1997**). In this manuscript, to have a common term between models and the data, we refer to sparse coding basis functions as RFs. Red – excitation, blue - inhibition. Inset shows axes. The full sets of visual spatial and corresponding spatiotemporal RFs for the sparse coding model when it is trained on visual inputs are shown in **Figure 5—figure supplements 1–2**. **Figure 5—figure supplements 3–5** show the auditory spectrotemporal and visual spatial and 2D spatiotemporal RFs of the sparse coding model when it was trained on inputs without added noise.

*Figure 5 continued on next page*

*Figure 5 continued*

DOI: https://doi.org/10.7554/eLife.31557.014

The following figure supplements are available for figure 5:

**Figure supplement 1.** Full set of visual spatial RFs of sparse coding model units.

DOI: https://doi.org/10.7554/eLife.31557.015

**Figure supplement 2.** 2D (space-time) visual spatiotemporal RFs of sparse coding model units.

DOI: https://doi.org/10.7554/eLife.31557.016

**Figure supplement 3.** Full set of auditory RFs of sparse coding model trained on auditory inputs without added noise.

DOI: https://doi.org/10.7554/eLife.31557.017

**Figure supplement 4.** Full set of visual spatial RFs of sparse coding model units trained on visual inputs without added noise.

DOI: https://doi.org/10.7554/eLife.31557.018

**Figure supplement 5.** 2D (space-time) visual spatiotemporal RFs of sparse coding model units trained on visual inputs without added noise.

DOI: https://doi.org/10.7554/eLife.31557.019

*4* and *6c–e*, and *Figure 4—figure supplement 1*). This is not the case with the sparse coding model, where units tend to show either excitation and inhibition having the same duration or an elongated temporal structure that does not show such stereotyped polarity changes (*Figures 5* and *6c–e*). It is also the case that the absolute timescales of excitation and inhibition match the data more closely in the case of the temporal prediction model (*Figure 6c–e*), although a few units display inhibition of a longer duration than is seen in the data (*Figure 6c*). The sparse coding model shows a wide range of temporal spans of excitation and inhibition, in keeping with previous studies (*Carlson et al., 2012*; *Carlin and Elhilali, 2013*).

Regarding the spectral properties of real neuronal RFs, the spans of inhibition and excitation over sound frequency tend to be similar (*Figure 6f–h*). This is also seen in the temporal prediction model, albeit with slightly more variation (*Figure 6f–h*). The sparse coding model shows more extensive variation in frequency spans than either the data or our model (*Figure 6f–h*).

## Quantitative analysis of visual results

We also compared the spatiotemporal RFs derived from the temporal prediction and sparse coding models with restricted published datasets summarizing RF characteristics of V1 neurons (*Ringach, 2002*) and a small number of full spatiotemporal visual RFs acquired from Ohzawa et al (*Ohzawa et al., 1996*). We assessed the orientation and spatial frequency tuning properties of the models' RFs by fitting Gabor functions to them (see Materials and methods).

We compared temporal properties of the RFs from the neural data and the temporal prediction model. In both cases, most power (mean over space and neurons of squared values) is in the most recent time steps (*Figure 7a*). Previous normative models of spatiotemporal RFs (*van Hateren and Ruderman, 1998a*; *Hyvärinen et al., 2003*; *Olshausen, 2003*) (*Figure 7—figure supplement 1c–d*) do not show this property, being either invariant over time or localized, but with a symmetric profile that is not restricted to the recent past. We also measured the space-time separability of the RFs of the temporal prediction model (see Materials and methods); substantial numbers of both space-time separable and inseparable units were apparent (631 separable, 969 inseparable; *Figure 4—figure supplement 3*). In addition to this, we measured the tilt direction index (TDI) of the model units from their 2D spatiotemporal RFs. This index indicates spatiotemporal asymmetry in space-time RFs and correlates with direction selectivity (*DeAngelis et al., 1993*; *Pack et al., 2006*; *Anzai et al., 2001*; *Baker, 2001*; *Livingstone and Conway, 2007*). The mean TDI for the population was 0.34 (0.29 SD), comparable with the ranges in the neural data (mean 0.16; 0.12 SD in cat area 17/18 (*Baker, 2001*), mean 0.51; 0.30 SD in macaque V1 [*Livingstone and Conway, 2007*]). Finally, we observed an inverse correlation ($r^2 = -0.33$, $p<10^{-9}$, n = 1205) between temporal and spatial frequency tuning (See Materials and methods), which is also a property of real V1 RFs (*DeAngelis et al., 1993*) and is seen in a sparse-coding-related model (*van Hateren and Ruderman, 1998a*).

The spatial tuning characteristics of the temporal prediction model's RFs displayed a wide range of orientation and spatial frequency preferences, consistent with the neural data (*DeAngelis et al., 1993*; *Kreile et al., 2011*) (*Figure 4—figure supplement 2*). Both model and real RFs (*Kreile et al., 2011*) show a preference for spatial orientations along the horizontal and vertical axes, although this

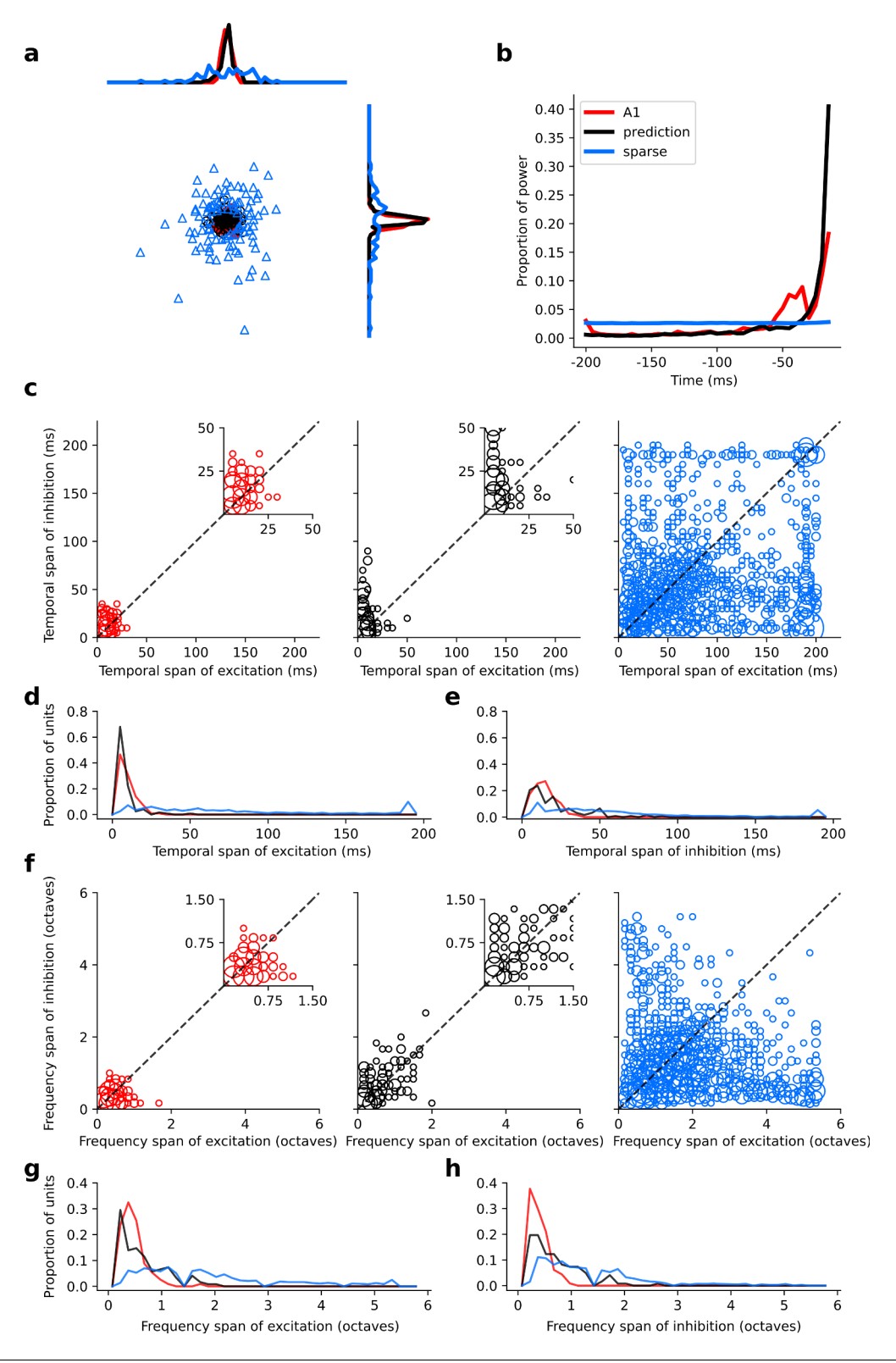

**Figure 6.** Population measures for real A1, temporal prediction model and sparse coding model auditory spectrotemporal RFs. The population measures are taken from the RFs shown in *Figures 3–5*. (**a**), Each point represents a single RF (with 32 frequency and 38 time steps) which has been embedded in a 2-dimensional space using Multi-Dimensional Scaling (MDS). Red circles - real A1 neurons, black circles – temporal prediction model units, blue triangles – sparse coding model units. Colour scheme applies to all subsequent panels. (**b**), Proportion of power contained in each time step

*Figure 6 continued on next page*

*Figure 6 continued*

of the RF, taken as an average across the population of units. (**c**), Temporal span of excitatory subfields versus that of inhibitory subfields, for real neurons and temporal prediction and sparse coding model units. The area of each circle is proportional to the number of occurrences at that point. The inset plots, which zoom in on the distribution use a smaller constant of proportionality for the circles to make the distributions clearer. (**d**), Distribution of temporal spans of excitatory subfields, taken by summing along the x-axis in (**c**). (**e**), Distribution of temporal spans of inhibitory subfields, taken by summing along the y-axis in (**c**). (**f**), Frequency span of excitatory subfields versus that of inhibitory subfields, for real neurons and temporal prediction and sparse coding model units. (**g**), Distribution of frequency spans of excitatory subfields, taken by summing along the x-axis in (**f**). (**h**), Distribution of frequency spans of inhibitory subfields, taken by summing along the y-axis in (**f**). *Figure 6—figure supplement 1* shows the same analysis for the temporal prediction model and sparse coding model trained on auditory inputs without added noise.

DOI: https://doi.org/10.7554/eLife.31557.020

The following figure supplement is available for figure 6:

**Figure supplement 1.** Population measures for real A1, temporal prediction model and sparse coding model auditory spectrotemporal RFs when models are trained on auditory inputs without added noise.

DOI: https://doi.org/10.7554/eLife.31557.021

orientation bias is seen to a greater extent in the temporal prediction model than in the data. The orientation and frequency tuning characteristics are also well captured by sparse coding related models of spatiotemporal RFs (*van Hateren and Ruderman, 1998a*; *Olshausen, 2003*) (*Figure 7—figure supplement 1e-f*). Furthermore, the widths and lengths of the RFs of the temporal prediction model, relative to the period of their oscillation, also match the neural data well (*Figure 7d*). The distribution of units extends along a curve from blob-like RFs, which lie close to the origin in this plot, to stretched RFs with several subfields, which lie further from the origin. Although this property is again fairly well captured by previous models (*Olshausen and Field, 1996*, *Olshausen and Field, 1997*; *Berkes et al., 2009*; *Ringach, 2002*; *van Hateren and van der Schaaf, 1998b*) (*Figure 7—figure supplement 1g*), only the temporal prediction model seems to be able to capture the blob-like RFs that form a sizeable proportion of the neural data (*Ringach, 2002*) (*Figure 7d* where $n_x$ and $n_y < ~0.25$, *Figure 4—figure supplement 2*). A small proportion of the population have RFs with several short subfields, forming a wing from the main curve in *Figure 7d*.

## Optimizing predictive capacity

Under our hypothesis of temporal prediction, we would expect that the better the temporal prediction model network is at predicting the future, the more the RFs of the network should resemble those of real neurons. To examine this hypothesis, we plotted the prediction error of the network as a function of two hyperparameters; the regularization strength and the number of hidden units (*Figure 8a*). Then, we plotted the similarity between the auditory RFs of real A1 neurons and those of the temporal prediction model (*Figure 8b*), as measured by the mean KS distances of the temporal and frequency span distributions (*Figure 6d–e,g–h*, Materials and methods). The set of hyperparameter settings that give good predictions are also those where the temporal prediction model produces RFs that are most similar to those recorded in A1 ($r^2 = 0.8$, $p<10^{-9}$, $n = 55$). This result argues that cortical neurons are indeed optimized for temporal prediction.

When the similarity measure was examined as a function of the same hyperparameters for the sparse coding model (*Figure 8—figure supplement 1*), and this was compared to that model's stimulus reconstruction capacity as a function of the same hyperparameters, a monotonic relationship between stimulus reconstruction capacity and similarity of real RFs was not found (*Figure 8—figure supplement 1*; $r^2 = -0.05$, $p=0.69$, $n = 50$). In previous studies in which comparisons have been made between normative models and real data, the model hyperparameters have been selected to maximize the similarity between the real and model RFs. In contrast, the temporal prediction model provides an independent criterion, the prediction error, to perform hyperparameter selection. To our knowledge, no such effective, measurable, independent criterion for hyperparameter selection has been proposed for other normative models of RFs.

## Variants of the temporal prediction model

The change in the qualitative structure of the RFs as a function of the number of hidden units and $L_1$ regularization strength, for both the visual and auditory models, can be seen in the interactive supplementary figures (*Figure 8—figure supplements 2–3*; https://yossing.github.io/temporal_

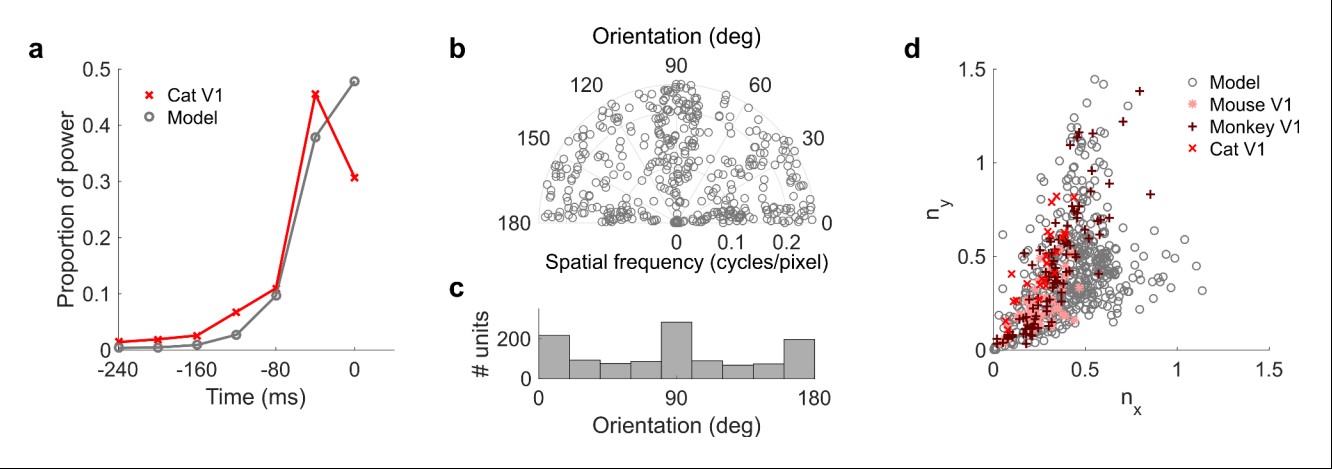

**Figure 7.** Population measures for real V1 and temporal prediction model visual spatial and spatiotemporal RFs. Model units were obtained by training the model with 1600 hidden units on visual inputs. The hidden unit number and $L_1$ weight regularization strength ($10^{-6.25}$) was chosen because it results in the lowest MSE on the prediction task, as measured using a cross validation set. Example units in *Figure 2* come from this set. (a), Proportion of power (sum of squared weights over space and averaged across units) in each time step, for real (*Ohzawa et al., 1996*) and model populations. (b), Joint distribution of spatial frequency and orientation tuning for population of model unit RFs at their time step with greatest power. (c), Distribution of orientation tuning for population of model unit RFs at their time step with greatest power. (d), Distribution of RF shapes for real neurons (cat, *Jones and Palmer, 1987*, mouse, *Niell and Stryker, 2008* and monkey, *Ringach, 2002*) and model units. $n_x$ and $n_y$ measure RF span parallel and orthogonal to orientation tuning, as a proportion of spatial oscillation period (*Ringach, 2002*). For (b–d), only units that could be well approximated by Gabor functions (n = 1205 units; see Materials and methods) were included in the analysis. Of these, only model units that were space-time separable (n = 473) are shown in (d) to be comparable with the neuronal data (*Ringach, 2002*). A further 4 units with $1.5 < n_y < 3.1$ are not shown in (d). *Figure 7—figure supplements 1–3* show example visual RFs and the same population measures for the sparse coding model trained on visual inputs with added noise and for the temporal prediction and sparse coding models trained on visual inputs without added noise.

DOI: https://doi.org/10.7554/eLife.31557.022

The following figure supplements are available for figure 7:

**Figure supplement 1.** Visual RFs and population measures for real V1 neurons and sparse coding model units.
DOI: https://doi.org/10.7554/eLife.31557.023

**Figure supplement 2.** Visual RFs and population measures for real V1 neurons and temporal prediction model units trained on visual inputs without added noise.
DOI: https://doi.org/10.7554/eLife.31557.024

**Figure supplement 3.** Visual RFs and population measures for real V1 neurons and sparse coding model units trained on visual inputs without added noise.
DOI: https://doi.org/10.7554/eLife.31557.025

prediction_model/figures/interactive_supplementary_figures.html) The main effect of the regularization is to restrict the RFs in space for the visual case and in frequency and time for the auditory case. When the regularization is non-existent or substantially weaker than the optimum for prediction, the visual RFs become less localized in space with more elongated bars. The auditory RFs become more disordered, losing clear structure in most cases. When the regularization is made stronger than the optimum, the RFs become more punctate, for both the visual and auditory models. When the regularization strength is at the optimum for prediction, the auditory and visual model RFs qualitatively most closely resemble those of A1 neurons and V1 simple cells, respectively. This is consistent with what we found quantitatively in the previous section for the auditory model.

The temporal prediction model and the sparse coding model both produce oriented Gabor-like RFs when trained on visual inputs. This raises the possibility that optimization for prediction implicitly optimizes for a sparse response distribution, and hence leads to oriented RFs. To test for this, we measured the sparsity of the visual temporal prediction model's hidden unit activities (by the Vinje-Gallant measure [*Baker, 2001*]) in response to the natural image validation set. Examining the relationship between predictive capacity and sparsity, over the range of $L_1$ weight regularization strength and hidden units explored, we did not find a clear monotonic relationship. Indeed, in both the auditory and visual cases, the hidden unit and $L_1$ regularization combination with the best

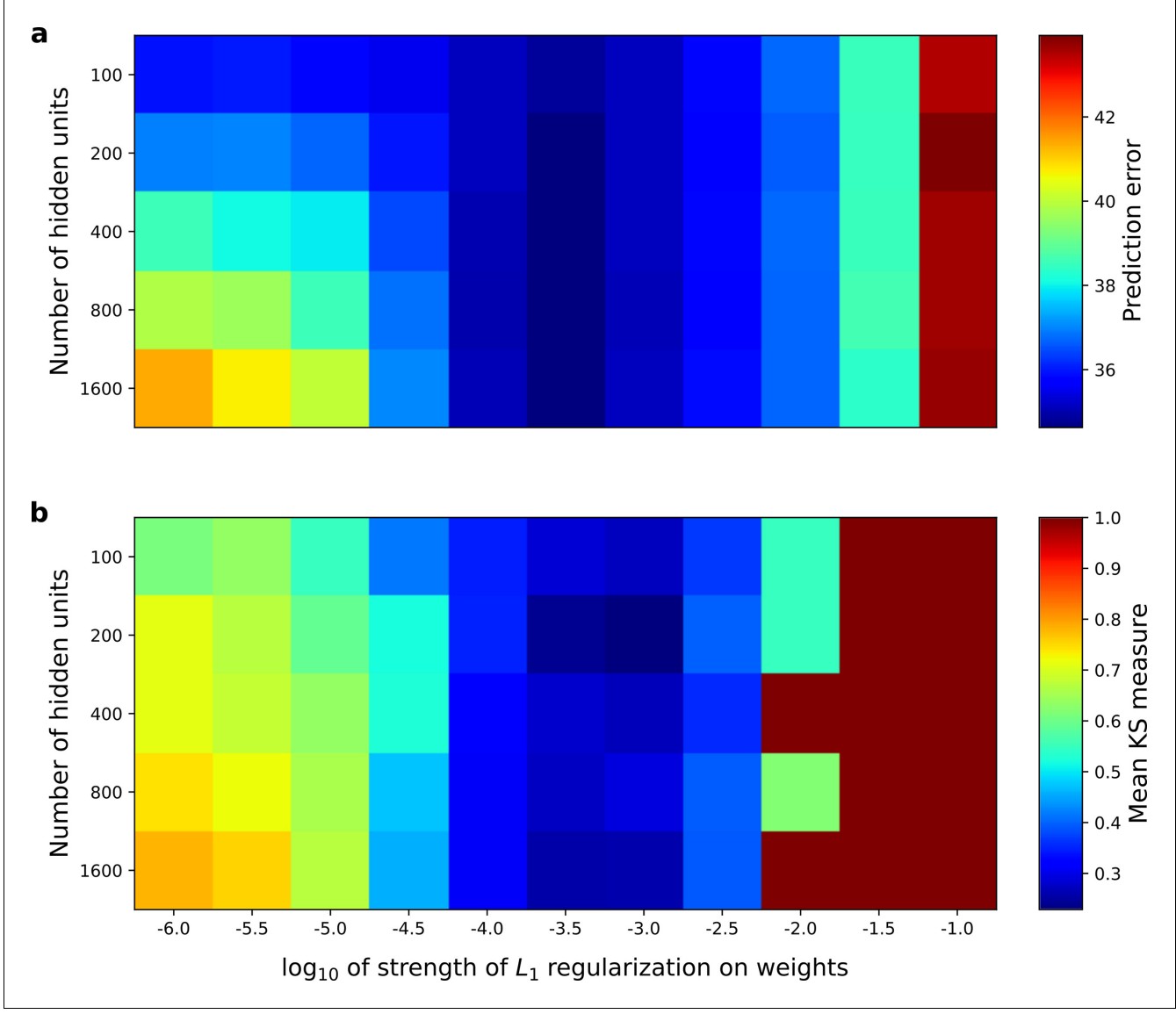

**Figure 8.** Correspondence between the temporal prediction model's ability to predict future auditory input and the similarity of its units' responses to those of real A1 neurons. Performance of model as a function of number of hidden units and $L_1$ regularization strength on the weights as measured by (a), prediction error (mean squared error) on the validation set at the end of training and (b), similarity between model units and real A1 neurons. The similarity between the real and model units is measured by averaging the Kolmogorov-Smirnov distance between each of the real and model distributions for the span of temporal and frequency tuning of the excitatory and inhibitory RF subfields (e.g. the distributions in *Figure 6d–e* and *Figure 6g–h*). *Figure 8—figure supplement 1* shows the same analysis, performed for the sparse coding model, which does not produce a similar correspondence.

DOI: https://doi.org/10.7554/eLife.31557.026

The following figure supplements are available for figure 8:

**Figure supplement 1.** Correspondence between sparse coding model's ability to reproduce its input and the similarity of its units' responses to those of real A1 neurons.

DOI: https://doi.org/10.7554/eLife.31557.027

**Figure supplement 2.** Interactive figure exploring the relationship between the strength of $L_1$ regularization on the network weights and the structure of the RFs the network produces when the network is trained on auditory inputs.

DOI: https://doi.org/10.7554/eLife.31557.028

**Figure supplement 3.** Interactive figure exploring the relationship between the strength of $L_1$ regularization on the network weights and the structure of the RFs the network produces when the network is trained on visual inputs.

DOI: https://doi.org/10.7554/eLife.31557.029

prediction had intermediate sparsity. For the visual case, the best-predicting model had sparsity 0.25, and other models within the grid search had sparsity ranging from 0.16 to 0.57. For the auditory case, the best-predicting model had sparsity 0.58, and other models had sparsity ranging from 0.42 to 0.69.

We also varied other characteristics of the temporal prediction model to understand their influence. For both the auditory and visual models, when a different hidden unit nonlinearity (tanh or rectified linear) was used, the networks had similar predictive capacity and produced comparable RFs. However, when the temporal prediction model had linear hidden units, it no longer predicted as well and produced RFs that were less like real neurons in their structure. For the auditory model, the linear model RFs generally became more narrowband in frequency with temporally extended excitation, instead of extended lagging inhibition (*Figure 4—figure supplement 4*). For the visual model, the linear model RFs also showed substantially less similarity to the V1 data. At low regularization (the best predicting case), the RFs formed full-field grid-like structures. At higher regularization, they were more punctate, with some units having oriented RFs with short subfields. The RFs also did not change form or polarity over time, but simply decayed into the past.

The temporal prediction model and sparse coding model results shown in the main figures of this paper were trained on inputs with added Gaussian noise (6 dB SNR), mimicking inherent noise in the nervous system. To determine the effect of adding this noise, all models were also trained without noise, producing similar results (*Figure 4—figure supplements 5–7*; *Figure 5—figure supplements 3–5*; *Figure 6—figure supplement 1*; *Figure 7—figure supplements 2–3*). The results were also robust to changes in the duration of the temporal window being predicted. We trained the auditory model to predict a span of either 1, 3, 6, or 9 time steps into the future and the visual model to predict 1, 3 or 6 time steps into the future. For the auditory case, we found that increasing the number of time steps being predicted had little effect on the RF structure, both qualitatively and by the KS measure of similarity to the real data. In the visual case, Gabor-like units were present in all cases. Increasing the number of time steps made the RFs more restricted in space and increased the proportion of blob-like RFs.

## Discussion

We hypothesized that finding features that can efficiently predict future input from its past is a principle that influences the structure of sensory RFs. We implemented an artificial neural network model that instantiates a restricted version of this hypothesis. When this model was trained using natural sounds, it produced RFs that are both qualitatively and quantitatively similar to those of A1 neurons. Similarly, when we trained the model using natural movies it produced RFs with many of the properties of V1 simple cells. This similarity is particularly notable in the temporal domain; the model RFs have asymmetric envelopes, with a preference for the very recent past, as is seen in A1 and V1. Finally, the more accurate a temporal prediction model is at prediction, the more its RFs tend to be like real neuronal RFs by the measures we use for comparison.

### Relationship to other models

A number of principles, often acting together, have been proposed to explain the form and diversity of sensory RFs. These include efficient coding (*Barlow, 1959*; *Olshausen and Field, 1996*, *Olshausen and Field, 1997*; *Carlson et al., 2012*; *Zhao and Zhaoping, 2011*; *Srinivasan et al., 1982*; *Brito and Gerstner, 2016*; *Olshausen, 2003*; *Attneave, 1954*), sparseness (*Olshausen and Field, 1996*, *Olshausen and Field, 1997*; *Carlson et al., 2012*; *Kozlov and Gentner, 2016*; *Brito and Gerstner, 2016*; *Olshausen, 2003*), and slowness (*Hyvärinen et al., 2003*; *Carlin and Elhilali, 2013*). Efficient coding indicates that neurons should encode maximal information about sensory input given certain constraints, such as spike count or energy costs. Sparseness posits that only a small proportion of neurons in the population should be active for a given input. Finally, slowness means that neurons should be sensitive to features that change slowly over time. The temporal prediction principle we describe here provides another unsupervised objective of sensory coding. It has been described in a very general manner by the information bottleneck concept (*Bialek et al., 2001*; *Salisbury and Palmer, 2016*; *Palmer et al., 2015*). We have instantiated a specific version of this idea, with linear-nonlinear encoding of the input, followed by a linear transform from the encoding units' output to the prediction.

In the following discussion, we describe previous normative models that infer RFs with temporal structure from auditory or movie input and relate them to spectrotemporal RFs in A1 or simple cell spatiotemporal RFs in V1, respectively. For focus, other normative models of less directly relevant areas, such as spatial receptive fields without a temporal component (*Olshausen and Field, 1996*, *Olshausen and Field, 1997*), complex cells (*Berkes and Wiskott, 2005*), retinal receptive fields (*Huang and Rao, 2011*; *Srinivasan et al., 1982*), or auditory nerve impulse responses (*Smith and Lewicki, 2006*), will not be examined.

## Auditory normative models

A number of coding objectives have been explored in normative models of A1 spectrotemporal RFs. One approach (*Zhao and Zhaoping, 2011*) found analytically that the optimal typical spectrotemporal RF for efficient coding was spectrally localized with lagging and flanking inhibition, and showed an asymmetric temporal envelope. However, the resulting RF also showed substantially more flanking inhibition, more ringing over time and frequency, and operated over a much shorter timescale (~10 ms) than seen in A1 RFs (*Figure 3*). Moreover, this approach produced a single generic RF, rather than capturing the diversity of the population.

Other models have produced a diverse range of spectrotemporal RFs. In the sparse coding approach (*Carlson et al., 2012*; *Brito and Gerstner, 2016*; *Młynarski and McDermott, 2017*; *Blättler et al., 2011*), a spectrogram snippet is reconstructed from a sum of basis functions (a linear generative model), each weighted by its unit's activity, with a constraint to have few active units. This approach is the same as the sparse coding model we used as a control (*Figure 5*). A challenge with many sparse generative models is that the activity of the units is found by a recurrent iterative process that needs to find a steady state; this is fine for static stimuli such as images, but for dynamic stimuli like sounds it is questionable whether the nervous system would have sufficient time to settle on appropriate activities before the stimulus had changed. Related work also used a sparsity objective, but rather than minimizing stimulus reconstruction error, forced high dispersal (*Kozlov and Gentner, 2016*) or decorrelation (*Klein et al., 2003*; *Carlin and Elhilali, 2013*) of neural responses. Although lacking some of the useful probabilistic interpretations of sparse generative models, this approach does not require a settling process for inference. An alternative to sparseness is temporal slowness, which can be measured by temporal coherence (*Carlin and Elhilali, 2013*). Here the linear transform from sequential spectrogram snippets to unit activity is optimized to maximize the correlation of each unit's response over a certain time window, while maintaining decorrelation between the units' activities.

Although the frequency tuning derived with these models can resemble that found in the midbrain or cortex (*Klein et al., 2003*; *Carlson et al., 2012*; *Kozlov and Gentner, 2016*; *Carlin and Elhilali, 2013*; *Brito and Gerstner, 2016*; *Młynarski and McDermott, 2017*; *Blättler et al., 2011*) (*Figure 5*), the resulting RFs lack the distinct asymmetric temporal profile and lagging inhibition seen in real midbrain or A1 RFs. Furthermore, they often have envelopes that are too elongated over time, often spanning the full temporal width of the spectrotemporal RF. This is related to the fact that the time window to be encoded by the model is set arbitrarily, and every time point within that window is given equal importance, that is, the direction of time is not accounted for. This is in contrast to the temporal prediction model, which naturally gives greater weighting to time-points near the present than to those in the past due to their greater predictive capacity.

## Visual normative models

The earliest normative model of spatiotemporal RFs of simple cells used independent component analysis (ICA) (*van Hateren and Ruderman, 1998a*), which is practically equivalent for visual or auditory data to the critically complete case of the sparse coding model (*Olshausen and Field, 1996*, *Olshausen and Field, 1997*) we used as a control (*Figure 5—figure supplements 1–2* and *Figure 7—figure supplement 1*). The RFs produced by this model and the control model reproduced fairly well the spatial aspects of simple cell RFs. However, in contrast to the temporal prediction model (*Figure 7d*), the subset of more 'blob-like' RFs seen in the data are not well captured by our control sparse coding model (*Figure 7—figure supplement 1g*). In the temporal domain, again unlike the temporal prediction model and real V1 simple cells, the RFs of the ICA and sparse coding models are not pressed up against the present with an asymmetrical temporal envelope, but instead

show a symmetrical envelope or span the entire range of times examined. A related model (*Olshausen, 2003*) assumes that a longer sequence of frames is generated by convolving each basis function with a time-varying sparse coefficient and summing the result, so that each basis function is applied at each point in time. The resulting spatiotemporal RFs are similar to those produced by ICA (*van Hateren and Ruderman, 1998a*), or our control model (*Figure 5—figure supplement 2* and *Figure 7—figure supplement 1c*). Although they tend not to span the entire range of times examined, they do show a symmetrical envelope, and require an iterative inference procedure, as described above for audition.

Temporal slowness constraints have also been used to model the spatiotemporal RFs of simple cells. The bubbles (*Hyvärinen et al., 2003*) approach combines sparse and temporal coherence constraints with reconstruction. The resulting RFs show similar spatial and temporal properties to those found using ICA. A related framework is slow feature analysis (SFA) (*Berkes and Wiskott, 2005*; *Wiskott and Sejnowski, 2002*), which enforces temporal smoothness by minimizing the derivative of unit responses over time, while maximizing decorrelation between units. SFA has been used to model complex cell spatiotemporal RFs (over only two time steps, *Berkes and Wiskott, 2005*), and a modified version has been used to model spatial (not spatiotemporal) RFs of simple cells (*Berkes et al., 2009*). These results are not directly comparable with our results or the spatiotemporal RFs of simple cells.

In the slowness framework, the features found are those that persist over time; the presence of such a feature in the recent past predicts that the same feature will be present in the near future. This is also the case for our predictive approach, which, additionally, can capture features in the past that predict features in the future that are subtly or radically different from themselves. The temporal prediction principle will also give different weighting to features, as it values predictive capacity rather than temporal slowness (*Creutzig and Sprekeler, 2008*). In addition, although slowness models can be extended to model RFs over more than one time step (*Berkes and Wiskott, 2005*; *Hyvärinen et al., 2003*; *Carlin and Elhilali, 2013*), capturing temporal structure, they do not inherently give more weighting to information in the most recent past and therefore do not give rise to asymmetric temporal profiles in RFs.

There is one study that has directly examined temporal prediction as an objective for visual RFs in a manner similar to ours (*Palm, 2012*). Here, as in our model, a single hidden layer feedforward neural network was used to predict the immediate future frame of a movie patch from its past frames. However, only two frames of the past were used in this study, so a detailed exploration of the temporal profile of the spatiotemporal RFs was not possible. Nevertheless, some similarities and differences in the spatial RFs between the two frames were noted, and some units had oriented RFs. In contrast to our model, however, many RFs were noisy and did not resemble those of simple cells. Potential reasons for this difference include the use of $L_2$ rather than $L_1$ regularization on the weights, an output nonlinearity not present in our model, the optimization algorithm used, network size, or the dataset. Another very recent related study (*Chalk et al., 2018*) also implemented a somewhat different form of temporal prediction, with a linear (rather than linear-nonlinear) encoder, and linear decoder. When applied to visual scenes, oriented receptive fields were produced, but they were spatio-temporally separable and hence not direction selective.

## Strengths and limitations of the temporal prediction model

Temporal prediction has several strengths as an objective function for sensory processing. First, it can capture underlying features in the world (*Bialek et al., 2001*); this is also the case with sparseness (*Olshausen and Field, 1996*, *Olshausen and Field, 1997*) and slowness (*Wiskott and Sejnowski, 2002*), but temporal prediction will prioritize different features. Second, it can predict future inputs, which is very important for guiding action, especially given internal processing delays. Third, objectives such as efficient or sparse reconstruction retain everything about the stimulus, whereas an important part of neural information processing is the selective elimination of irrelevant information (*Marzen and DeDeo, 2017*). Prediction provides a good initial criterion for eliminating potentially unwanted information. Fourth, prediction provides a natural method to determine the hyperparameters of the model (such as regularization strength, number of hidden units, activation function and temporal window size). Other models select their hyperparameters depending on what best reproduces the neural data, whereas we have an independent criterion – the capacity of the network to predict the future. One notable hyperparameter is how many time-steps of past input to encode.

As described above, this is naturally decided by our model because only time-steps that help predict the future have significant weighting. Fifth, the temporal prediction model computes neuronal activity without needing to settle to a steady state, unlike some other models (*Olshausen and Field, 1996*, *Olshausen and Field, 1997*; *Carlson et al., 2012*; *Brito and Gerstner, 2016*; *Młynarski and McDermott, 2017*). For dynamic stimuli, a model that requires settling may not reach equilibrium in time to be useful. Sixth, and most importantly, temporal prediction successfully models many aspects of the RFs of primary cortical neurons. In addition to accounting for spatial and spectral tuning in V1 and A1, respectively, at least as well as other normative models, it reproduces the temporal properties of RFs, particularly the asymmetry of the envelopes of RFs, something few previous models have attempted to explain.

Although the temporal prediction model's ability to describe neuronal RFs is high, the match with real neurons is not perfect. For example, the span of frequency tuning of our modelled auditory RFs is narrower than in A1 (*Figure 6g–h*). We also found an overrepresentation of vertical and horizontal orientations compared to real V1 data (*Figure 7b–c*). Some of these differences could be a consequence of the data used to train the model. Although the statistics of natural stimuli are broadly conserved (*Field, 1987*), there is still variation (*Torralba and Oliva, 2003*), and the dataset used to train the network may not match the sensory world of the animal experienced during development and over the course of evolution. In future work, it would be valuable to explore the influence of natural datasets with different statistics, and also to match those datasets more precisely to the evolutionary context and individual experience of the animals examined. Furthermore, a comparison of the model with neural data from different species, at different ages, and reared in different environments would be useful.

Another cause of differences between the model and neural RFs may be the recording location of the RFs and how they are characterized. We used the primary sensory cortices as regions for comparison, because we performed transformations on the input data that are similar to the preprocessing that takes place in afferent subcortical structures. We spatially filtered the visual data in a similar way to the retina (*Olshausen and Field, 1996*, *Olshausen and Field, 1997*), and spectrally decomposed the auditory data as in the inner ear, and then used time bins (5 ms) which are coarser than, but close to, the maximum amplitude modulation period that can be tracked by auditory midbrain neurons (*Rees and Møller, 1983*). However, primary cortex is not a homogenous structure, with neurons in different layers displaying certain differences in their response properties (*Harris and Mrsic-Flogel, 2013*). Furthermore, the methods by which neurons are sampled from the cortex may not provide a representative sample. For example, multi-electrode arrays tend to favour larger and more active neurons. In addition, the method and stimuli used to construct RFs from the data can bias their structure somewhat (*Willmore et al., 2016*).

The model presented here is based on a simple feedforward network with one layer of hidden units. This limits its ability to predict features of the future input, and to account for RFs with nonlinear tuning. More complex networks, with additional layers or recurrency may allow the model to account for more complex tuning properties, including those found beyond the primary sensory cortices. Careful, principled adjustment of the preprocessing, or different regularization methods (such as sparseness or slowness applied to the units' activities), may also help. There is an open question as to whether the current model may eliminate some information that is useful for reconstruction of the past input or for prediction of higher order statistical properties of the future input, which might bring it into conflict with the principle of least commitment (*Marr, 1976*). It is an empirical question how much organisms preserve information that is not predictive of the future, although there are theoretical arguments against such preservation (*Bialek et al., 2001*). Such conflict might be remedied, and the model improved, by adding feedback from higher areas or by adding an objective to reconstruct the past or present (*Barlow, 1959*; *Olshausen and Field, 1996*, *Olshausen and Field, 1997*; *Attneave, 1954*) in addition to predicting the future.

To determine whether the model could help explain neuronal responses in higher areas, it would be useful to develop a hierarchical version of the temporal prediction model, applying the same model again to the activity of the hidden units rather than to the input. Another useful extension would be to see if the features learnt by the temporal prediction model could be used to accelerate learning of useful tasks such as speech or object recognition, by providing input or initialization for a supervised or reinforcement learning network. Indeed, temporal predictive principles have been

shown to be useful for unsupervised training of networks used in visual object recognition (*Srivastava et al., 2015*; *Ranzato, 2016*; *Lotter et al., 2016*; *Oh et al., 2015*).

Finally, it is interesting to consider possible more explicit biological bases for our model. We envisage the input units of the model as thalamic input, and the hidden units as primary cortical neurons. Although the function of the output units could be seen as just a method to optimize the hidden units to find the most predictive code given sensory input statistics, they may also have a physiological analogue. Current evidence (*Dahmen and King, 2007*; *Huberman et al., 2008*; *Kiorpes, 2015*) suggests that while primary cortical RFs are to an extent hard-wired in form by natural selection, their tuning is also refined by individual sensory experience. This refinement process may require a predictive learning mechanism in the animal's brain, at least at some stage of development and perhaps also into adulthood. Hence, one might expect to find a subpopulation of neurons that represent the prediction (analogous to the output units of the model) or the prediction error (analogous to the difference between the output unit activity and the target). Indeed, signals relating to sensory prediction error have been found in A1 (*Rubin et al., 2016*), though they may also be located in other regions of the brain. Finally, it is important to note that, although the biological plausibility of backpropagation has long been questioned, recent progress has been made in developing trainable networks that perform similarly to artificial neural networks trained with backpropagation, but with more biologically plausible characteristics (*Bengio et al., 2015*), for example, by having spikes or avoiding the weight transport problem (*Lillicrap et al., 2016*).

## Conclusion

We have shown that a simple principle - predicting the imminent future of a sensory scene from its recent past - explains many features of the RFs of neurons in both primary visual and auditory cortex. This principle may also account for neural tuning in other sensory systems, and may prove useful for the study of higher sensory processing and aspects of neural development and learning. While the importance of temporal prediction is increasingly widely recognized, it is perhaps surprising nonetheless that many basic tuning properties of sensory neurons, which we have known about for decades, appear, in fact, to be a direct consequence of the brain's need to efficiently predict what will happen next.

# Materials and methods

## Data used for model training and testing

### Visual inputs

Videos (without sound, sampled at 25 fps) of wildlife in natural settings were used to create visual stimuli for training the artificial neural network. The videos were obtained from http://www.arkive. org/species, contributed by: BBC Natural History Unit, http://www.gettyimages.co.uk/footage/ bbcmotiongallery; BBC Natural History Unit and Discovery Communications Inc., http://www.bbcmotiongallery.com; Granada Wild, http://www.itnsource.com; Mark Deeble and Victoria Stone, Flat Dog Productions Ltd., http://www.deeblestone.com; Getty Images, http://www.gettyimages.com; National Geographic Digital Motion, http://www.ngdigitalmotion.com. The longest dimension of each video frame was clipped to form a square image. Each frame was then band-pass filtered (*Olshausen and Field, 1997*) and downsampled (using bilinear interpolation) over space, to provide 180 × 180 pixel frames. Non-overlapping patches of 20 × 20 pixels were selected from a fixed region in the centre of the frames, where there tended to be visual motion. The video patches were cut into sequential overlapping clips each of 8 frames duration. Thus, each training example (clip) was made up of a 20 × 20 pixel section of the video with a duration of 8 frames (320 ms), providing a training set of $N$ =~500,000 clips from around 5.5 hr of video, and a validation set of $N$ =~100,000 clips. Finally, the training and validation sets were normalized by subtracting the mean and dividing by the standard deviation (over all pixels, frames and clips in the training set). The goal of the neural network was to predict the final frame (the 'future') of each clip from the first seven frames (the 'past').

## Auditory inputs

Auditory stimuli were compiled from databases of human speech (~60%), animal vocalizations (~20%) and sounds from inanimate objects found in natural settings (e.g. running water, rustling leaves; ~20%). Stimuli were recorded using a Zoom H4 or collected from online sources. Natural sounds were obtained from www.freesound.org, contributed by users sedi, higginsdj, jult, kvgarlic, xenognosis, zabuhailo, funnyman374, videog, j-zazvurek, samueljustice00, gfrog, ikbenraar, felix-blume, orbitalchiller, saint-sinner, carlvus, vflefevre, hitrison, willstepp, timbahrij, xdimebagx, r-nd0mm3m, the-yura, rsilveira-88, stomachache, foongaz, edufigg, yurkobb, sandermotions, darius-kedros, freesoundjon-01, dwightsabeast, borralbi, acclivity, J.Zazvurek, Zabuhailo, soundmary, Darius Kedros, Kyster, urupin, RSilveira and freelibras. Human speech sounds were obtained from http://databases.forensic-voice-comparison.net/ (*Morrison et al., 2015*, *Morrison et al., 2012*).

Each sound was sampled at (or resampled to) 44.1 kHz and converted into a simple 'cochleagram', to make it more analogous to the activity pattern that would be passed to the auditory pathway after processing by the cochlea. To calculate the cochleagram, a power spectrogram was computed using 10 ms Hamming windows, overlapping by 5 ms (giving time steps of 5 ms). The power across neighbouring Fourier frequency components was then aggregated into 32 frequency channels using triangular windows with a base width of 1/3 octave whose centre frequencies ranged from 500 to 17,827 Hz (1/6$^{th}$ octave spacing, using code adapted from melbank.m, http://www.ee.ic.ac.uk/hp/staff/dmb/voicebox/voicebox.html). The cochleagrams were then decomposed into sequential overlapping clips, each of 43 time steps (415 ms) in duration, providing a training set of ~1,000,000 clips (~1.3 hr of audio) and a validation set of ~200,000 clips. To approximately model the intensity compression seen in the auditory nerve (*Sachs and Abbas, 1974*), each frequency band in the stimulus set was divided by the median value in that frequency band over the training set, and passed through a hill function, defined as $h(x) = cx/(1 + cx)$ with $c$ = 0.02. Finally, the training and cross-validation sets were normalized by subtracting the mean and dividing by the standard deviation over all time steps, frequency bands and clips in the training set. The first 40 time steps (200 ms) of each clip (the 'past') were used as inputs to the neural network, whose aim was to predict the content (the 'future') of the remaining three time steps (15 ms).

## Addition of Gaussian noise

To replicate the effect of noise found in the nervous system, Gaussian noise was added to both the auditory and visual inputs with a signal-to-noise ratio (SNR) of 6 dB. While the addition of noise did not make substantial differences to the RFs of units trained on visual inputs, this improved the similarity to the data when the model was trained on auditory inputs. The results from training the network on inputs without added noise are shown for auditory inputs in *Figure 4—figure supplement 5* and *Figure 6—figure supplement 1* and for visual inputs in *Figure 4—figure supplements 6–7* and *Figure 7—figure supplement 2*. The results from the sparse coding model were similar in both cases for inputs with and without noise (*Figures 5–6*, *Figure 5—figure supplements 1–5*, *Figure 6—figure supplement 1*, *Figure 7—figure supplements 1* and *3*).

## Temporal prediction model

### The model and cost function

The temporal prediction model was implemented using a standard fully connected feed-forward neural network with one hidden layer. Each hidden unit in the network computed the linear weighted sum of inputs, and its output was determined by passing this sum through a monotonic nonlinearity. This nonlinearity $s = h(a)$ was either a logistic function $h(a) = 1/(1 + \exp(-a))$ or a similar nonlinear function (such as $\tanh$). For results reported here, we used the logistic function, though obtained similar results when we trained the model using $h(a) = \tanh(a)$. For comparison, we also trained the model replacing the nonlinearity with a linear function, where $h(a) = a$. In this case, we found that the RFs tended to be punctate in space or frequency and did not typically show the alternating excitation and inhibition over time that is characteristic real neurons in A1 and V1.

Formally, for a network with $i = 1$ to $I$ input variables, $k = 1$ to $K$ output units and a single layer of $j = 1$ to $J$ hidden units, the output $s_{jn}$ of hidden unit $j$ for clip $n$ is given by:

$$s_{jn} = \mathrm{h}\left( b_j + \sum_{i=1}^{I} w_{ji} u_{in} \right) \qquad (1)$$

The value $u_{in}$ of input variable $i$ for clip $n$ is simply the value for a particular pixel and time step (frame) of the 'past' in preprocessed visual clip $n$ ($I$ = 20 pixels × 20 pixels × 7 time steps = 2800), or the value for a particular frequency band and time step of the 'past' of cochleagram clip $n$ ($I$ = 32 frequencies × 40 time steps = 1280). Hence, the index $i$ spans over several frequencies or pixels and also over time steps into the past. The subscript $n$ has been dropped for clarity in the figures (**Figure 1**). The parameters in **Equation 1** are the connective input weights $w_{ji}$ (between each input variable $i$ and hidden unit $j$), and the bias $b_j$ (of hidden unit $j$).

The activity $\hat{v}_{kn}$ of each output unit $k$, which is the estimate of the true future $v_{kn}$ given the past $u_{in}$, is given by:

$$\hat{v}_{kn} = b_k + \sum_{j=1}^{J} w_{kj} s_{jn} \qquad (2)$$

The parameters in **Equation 2** are the connective output weights $w_{kj}$ (between each hidden unit $j$ and output unit $k$) and the bias $b_k$ (of output unit $k$). The activity $\hat{v}_{kn}$ of output unit $k$ for clip $n$ is the estimate for a particular pixel of the 'future' in the visual case ($K$ = 20 pixels × 20 pixels × 1 time step = 400), or the value for a particular frequency band and time step of the 'future' in the auditory case ($K$ = 32 frequencies × 3 time steps = 96).

The parameters $w_{ji}$, $w_{kj}$, $b_j$, and $b_k$ were optimized for the training set by minimizing the cost function given by:

$$E = \frac{1}{NK}\sum_{n=1}^{N}\sum_{k=1}^{K}(\hat{v}_{kn} - v_{kn})^2 + \lambda\left( \sum_{i=1}^{I}\sum_{j=1}^{J}|w_{ji}| + \sum_{j=1}^{J}\sum_{k=1}^{K}|w_{kj}| \right) \qquad (3)$$

Thus, $E$ is the mean squared error (the prediction error) between the prediction $\hat{v}_{kn}$ and the target $v_{kn}$ over all $N$ training examples and $K$ target variables, plus an $L_1$ regularization term, which is proportional to the sum of absolute values of all weights in the network and its strength is determined by the hyper-parameter $\lambda$. This regularization tends to drive redundant weights to near zero and provides a parsimonious network.

## Implementation details

The networks were implemented in Python (https://lasagne.readthedocs.io/en/latest/; http://deep-learning.net/software/theano/). The objective function was minimized using backpropagation as performed by the Adam optimization method (**Kingma and Adam, 2014**). An alternative implementation of the model was also made in MATLAB using the Sum-of-Functions Optimizer (**Sohl-Dickstein et al., 2014**) (https://github.com/Sohl-Dickstein/Sum-of-Functions-Optimizer) to train the network using backpropagation. Training examples were split into minibatches of approximately 200 training examples each.

During model network training, several hyperparameters were varied, including the regularization strength ($\lambda$), the number of units in the hidden layer and the nonlinearity used by each hidden unit. For each hyperparameter setting, the training algorithm was run for 1000 iterations. Running the network for longer (10000 iterations) showed negligible improvement to the prediction error (as measured on the validation set) or change in RF structure.

The effect of varying the number of hidden units and $\lambda$ on the prediction error for the validation set is shown in **Figure 8**. In both the visual and auditory case, the results presented (**Figure 2,4,6,7** and supplements) are the networks that predicted best on the validation set after 1000 iterations through the training data. For the auditory case, the settings that resulted in the best prediction were 1600 hidden units and $\lambda = 10^{-3.5}$, while in the visual case, the optimal settings were 1600 hidden units and $\lambda = 10^{-6.25}$.

## Model receptive fields

In the model, the combination of linear weights and nonlinear activation function are similar to the basic linear non-linear (LN) model (*Simoncelli et al., 2004*; *Dahmen et al., 2008*; *Atencio et al., 2008*; *Chichilnisky, 2001*; *Rabinowitz et al., 2011*) commonly used to describe neural RFs. Hence, the input weights between the input layer and a hidden unit of the model network are taken directly to represent the unit's RF, indicating the features of the input that are important to that unit.

Because of the symmetric nature of the sigmoid function, $h(a) = 1 - h(-a)$, after appropriate modification of the biases a hidden unit has the same influence on the prediction if its input and output matrices are both multiplied by $-1$. That is, for unit $j$, if we convert $w_{ij}$ to $-w_{ij}$, $w_{jk}$ to $-w_{jk}$, $b_j$ to $-b_j$, and $b_k$ to $-b_k + w_{jk}$, this will have no effect on the prediction or the cost function. This can be done independently for each hidden unit. Hence, the sign of each unit's RF could equally be positive or negative and have the same result on the predictions given by the network. However, we know that auditory units always have leading excitation (*Figure 3*). Hence, for both the predictive model and for the sparse coding model, we assume leading excitation for each unit. This was done for all auditory analyses.

As more units are added to the model network, the number of inactive units increases. To account for this, we measured the relative strength of all input connections to each hidden unit by summing the square of all input weights for that unit. Units for which the sum of square input weights was <1% of the maximum strength for the population were deemed to be inactive and excluded from all subsequent analyses. The difference in connection strength between active and inactive units was very distinct; a threshold <0.0001% only marginally increases the number of active units.

## Sparse coding model

The sparse coding model was used as a control for both visual and auditory cases. The Python implementation of this model (https://github.com/zayd/sparsenet) was trained using the same visual and auditory inputs used to train the predictive model. The training data were divided into mini-batches which were shuffled and the model optimized for one full pass through the data. Inference was performed using the Fast Iterative Shrinkage and Thresholding (FISTA) algorithm. A sparse $L_1$ prior with strength $\lambda$ was applied to the unit activities, providing activity regularization. A range of $\lambda$-values and unit numbers were tried (*Figure 8—figure supplement 1*). The learning rate and batch size were also varied until reasonable values were found. As there was no independent criterion by which to determine the 'best' settings, we chose the network that produced basis functions whose receptive fields were most similar to those of real neurons. In the auditory case, this was determined using the mean KS measure of similarity (*Figure 8—figure supplement 1*). In the visual case, as a similarity measure was not performed, this was done by inspection. In both cases, the model configurations chosen were restricted to those trained in an overcomplete condition (having more units than the number of input variables) in order to remain consistent with previous instantiations of this model (*Olshausen and Field, 1996*; *Olshausen and Field, 1997*; *Carlson et al., 2012*). In this manner, we selected a sparse coding network with 1600 units, $\lambda = 10^{0.5}$, learning rate = 0.01 and 100 mini-batches in the auditory case (*Figures 5–6*). In the visual case, the network selected was trained with 3200 units, $\lambda = 10^{0.5}$, learning rate = 0.05 and 100 mini-batches (*Figure 5—figure supplements 1–2* and *Figure 7—figure supplement 1*). Although the sparse coding basis functions are projective fields, they tend to be similar in structure to receptive fields (*Olshausen and Field, 1996*; *Olshausen and Field, 1997*), and, for simplicity, are referred to as RFs.

## Auditory receptive field analysis

### In vivo A1 RF data

Auditory RFs of neurons were recorded in the primary auditory cortex (A1) and anterior auditory field (AAF) of 5 pigmented ferrets of both sexes (all >6 months of age) and used as a basis for comparison with the RFs of model units trained on auditory stimuli. Systematic differences in response properties of A1 and AAF neurons are minor and not relevant for this study, and for simplicity here, we refer to neurons from either primary field indiscriminately as 'A1 neurons'. These recordings were performed under license from the UK Home Office and were approved by the University of Oxford Committee on Animal Care and Ethical Review. Full details of the recording methods are described

in earlier studies (*Willmore et al., 2016*; *Bizley et al., 2009*). Briefly, we induced general anaesthesia with a single intramuscular dose of medetomidine (0.022 mg · kg$^{-1}$ · h$^{-1}$) and ketamine (5 mg · kg$^{-1}$ · h$^{-1}$), which was then maintained with a continuous intravenous infusion of medetomidine and ketamine in saline. Oxygen was supplemented with a ventilator, and we monitored vital signs (body temperature, end-tidal $CO_2$, and the electrocardiogram) throughout the experiment. The temporal muscles were retracted, a head holder was secured to the skull surface, and a craniotomy and a durotomy were made over the auditory cortex. Extracellular recordings were made using silicon probe electrodes (Neuronexus Technologies) and acoustic stimuli were presented via Panasonic RPHV27 earphones, which were coupled to otoscope specula that were inserted into each ear canal, and driven by Tucker-Davis Technologies System III hardware (48 kHz sample rate).

The neuronal recordings used the 'BigNat' stimulus set (*Willmore et al., 2016*), which consists of natural sounds including animal vocalizations (e.g., ferrets and birds), environmental sounds (e.g., water and wind), and speech. To identify those neural units that were driven by the stimuli, we calculated a 'noise ratio' statistic (*Rabinowitz et al., 2011*; *Sahani and Linden, 2003*) for each unit and excluded from further analysis any units with a noise ratio >40. In total, driven spiking responses of 114 units (75 single unit, 39 multi-unit) were recorded to this stimulus set. Then, the auditory (spectrotemporal) RF of each unit was constructed using a previously described method (*Willmore et al., 2016*). Briefly, linear regression was performed in order to minimize the squared error between each neuron's spiking response over time and the cochleagram of the stimuli that gave rise to that response. The method used was exactly the same as in our earlier study (*Willmore et al., 2016*), except that $L_1$ rather than $L_2$ regularization was used to constrain the regression. The spectrotemporal RFs of these neurons took the same form as the inputs to the model neural network (i.e., 32 frequencies and 40 time-steps over the same range of values) and were therefore comparable to the model units' RFs. In order to account for the latency of auditory cortical responses, the most recent two time-steps (10 ms) of the neuronal RFs were removed, leaving 38 time-steps.

## Multi-dimensional scaling (MDS)

To get a non-parametric indication of how similar the model units' RFs were to those of real A1 neurons, each RF was embedded into a 2-dimensional space using MDS (*Figure 6a* and *Figure 6—figure supplement 1a*). First, 100 units each from the temporal prediction and sparse coding models and from the real population were chosen at random. To ensure that the model RFs were of the same dimensionality as the real RFs prior to embedding, the least recent two time steps of each model RF were removed.

## Measuring temporal and frequency spans of RFs

We quantified the span, over time and frequency, of the excitatory and inhibitory subfields of each RF. To do this, each RF was first separated into excitatory and inhibitory subfields, where the excitatory subfield was the RF with negative values set to 0, and the inhibitory subfield the RF with positive values set to 0. In some cases, model units did not exhibit notable inhibitory subfields. To account for this, the power contained in each subfield was calculated (sum of the squares of the subfield). Inhibitory subfields with <5% of the power of that unit's excitatory subfield were excluded from further analysis. According to this criterion, 44 of 167 active units in the temporal prediction model and 193 of 1600 units in the sparse model did not display inhibition.

Singular value decomposition (SVD) was performed on each subfield separately, and the first pair of singular vectors was taken, one of which is over time, the other over frequency. For the excitatory subfield, the temporal span was measured as the proportion of values in the temporal singular vector that exceeded 50% of the maximum value in the vector. The same analysis provided the temporal span for the inhibitory subfield. Similarly, we measured the frequency spans of the RFs by applying this measure to the frequency singular vectors of the excitatory and the inhibitory subfields.

We also examined, for both real and model RFs, the mean power for each of the 38 time steps in the RFs (*Figure 6b*), which was calculated as the mean of the squared RF values, over all frequencies and RFs, at each time step.

## Mean KS measure

To compare each network's units with those recorded in A1 (*Figure 3*), the two-sample Kolmogorov-Smirnov (KS) distance between the real and model distribution was measured for both the temporal and spectral span of the excitatory and inhibitory subfields (e.g. the distributions in *Figure 6d–e* and *Figure 6g–h*). These four KS measures were then averaged to give a single mean KS measure for each network, indicating how closely the temporal and frequency characteristics of real and model units matched on average for that network. The KS measure is low for similar distributions and high for distributions that diverge greatly. Thus networks whose units display temporal and frequency tuning characteristics that match those of real neurons more closely give rise to a lower mean KS measure.

## Visual receptive field analysis

### In vivo V1 RF data

Visual RFs measured using recordings from V1 simple cells were compared against the model (*Figure 2c*, and *Figure 7a*, cat, *Ohzawa et al., 1996*). The model was also compared to measures of simple cell RFs (*Figure 7d* and corresponding supplements, cat, *Jones and Palmer, 1987*, mouse, *Niell and Stryker, 2008* and monkey, *Ringach, 2002*). The data were taken from the authors' website (*Ringach, 2002*) or extracted from relevant papers (*Jones and Palmer, 1987*) or provided by the authors (*Ohzawa et al., 1996*; *Niell and Stryker, 2008*).

### Fitting Gabors

In order to quantify tuning properties of the model's visual RFs, 2D Gabors were fitted to the optimal time-step of each unit's response (*Jones and Palmer, 1987*; *Ringach, 2002*). This allowed comparison to previous experimental studies which parameterized real RFs by the same method (*Ringach, 2002*). The optimal time-step was defined (*Ringach, 2002*) as the time-step of the unit's response which contained the most power (mean square). The Gabor function has been shown to provide a good approximation for most spatial aspects of simple visual RFs (*Jones and Palmer, 1987*; *Ringach, 2002*). The 2D Gabor is given as:

$$G(x', y') = A \exp\left(-\left(\frac{x'}{\sqrt{2}\sigma_x}\right)^2 - \left(\frac{y'}{\sqrt{2}\sigma_y}\right)^2\right)\cos(2\pi f x' + \phi) \tag{4}$$

where, the spatial coordinates $(x', y')$ are acquired by translating the centre of the RF $(x_0, y_0)$ to the origin and rotating the RF by its spatial orientation $\theta$:

$$x' = (x - x_0)\cos\theta + (y - y_0)\sin\theta \tag{5}$$

$$y' = -(x - x_0)\sin\theta + (y - y_0)\cos\theta \tag{6}$$

$\sigma_x$ and $\sigma_y$ provide the width of the Gaussian envelope in the $x'$ and $y'$ directions, while $f$ and $\phi$ parameterize the spatial frequency and phase of the sinusoid along the $x'$ axis. $A$ parameterizes the height of the Gaussian envelope.

For each RF, the parameters $(x_0, y_0, \sigma_x, \sigma_y, \theta, f, \phi)$ of the Gabor were fitted by minimizing the mean squared error between the Gabor model and the RF using the minFunc minimization package (http://www.cs.ubc.ca/~schmidtm/Software/minFunc.html). In order to avoid local minima, the fitting was performed in two steps. First, the spatial RF was converted to the spectral domain using a 2D Fourier transform. Since the Fourier transform of a 2D Gabor is a 2D Gaussian (*Jones and Palmer, 1987*), which is easier to fit, an estimate of many of the parameters was obtained by first fitting a 2D Gaussian in the spectral magnitude domain. Using the parameters obtained from the spectral fitting as initial estimates, a 2D Gabor was then fitted to the original RF in the spatial domain. The fitted parameters provided a good estimate of the units' responses, with residual errors between the spatial responses and the corresponding Gabor fits being small and lacking spatial structure, and the median pixel-wise correlation coefficient of the Gabor fits for the temporal prediction model units was 0.88. Units whose fitted Gabors had a poor fit (those with a correlation coefficient <0.7; 214 units) were excluded from further analysis. We also excluded units with a high correlation coefficient (>0.7) if the centre position

of the Gabor was estimated to be outside the RF, and hence only the Gabor's tail was being fitted to the response (39 units), and those for which the estimated standard deviation of the Gaussian envelope in either x or y was <0.5 pixels, which meant very few non-negligible pixel values were used to constrain the parameters (146 units). Together, these exclusion criteria (which sometimes overlapped), led to 395 of the 1600 responsive units being excluded for the temporal prediction model.

## 2D spatiotemporal receptive fields

In order to better view their temporal characteristics we collapsed the 3D spatiotemporal real and model RFs (space-space-time) along a single spatial direction to create 2D spatiotemporal (space-time) representations (*DeAngelis et al., 1993*). First, we determined the 3D RFs' optimal time step (the time step with the largest sum of squared values). We then acquired the rotation and translation that centres the RF on zero and places the oriented bars parallel to the y-axis at the optimal time step from the Gabor parameterization of each unit at its optimal time step. We applied this fixed transformation to each time step and collapsed the RF by summing the activity along the newly defined y-axis. The resulting 2D (space-time) RFs provide intuitive visualization of the RF across time, while losing minimal information. For the RFs of real neurons (*Ohzawa et al., 1996*), the most recent time step (40 ms) of the 3D and 2D spatiotemporal RFs were removed to account for the latency of V1 neurons (*Figures 2c and 7a*).

## Estimating space-time separability

The population of model units contained both space-time (ST) separable and inseparable units. First the two spatial dimensions of the $20 \times 20 \times 7$ 3D RF were collapsed to a single vector to yield a single $400 \times 7$ matrix. The SVD of this matrix was then taken and the singular values examined. If the ratio between the second and first singular value was $\geq 0.5$, the unit was deemed to be inseparable. Otherwise, the unit was deemed to be separable. Examining the $20 \times 7$ 2D spatiotemporal RFs (obtained as outlined in the preceding section; *Figure 4—figure supplement 3*) showed this to be an accurate way of separating space-time separable and inseparable units.

## Spatial RF structure

For comparison with the real V1 RF and previous theoretical studies, the width and length of our model's RFs were measured relative to their spatial frequency (*Ringach, 2002*). Here, $n_y = \sigma_y f$ gives a measure of the length of the bars in the RF, while $n_x = \sigma_x f$ gives a measure of the number of oscillations of its sinusoidal component. Thus, in the $n_y$, $n_x$ plane, blob-like RFs with few cycles lie close to the origin, while stretched RFs with many subfields lie away from the origin. RFs with values high along the $n_x$ axis, have many bars, while those far along the $n_y$ axis have long bars. As in Ringach (*Ringach, 2002*) only space-time separable units were included in this analysis.

## Temporal weighting profile of the population

The mean power for each of the seven time steps of the RFs was examined for both real and model populations (*Figure 7a*). The temporal weighting profile was calculated as the mean, over space and the population, of the squared values of the 2D spatiotemporal RFs at each time step.

## Tilt direction index

The tilt direction index (TDI) (*DeAngelis et al., 1993*; *Pack et al., 2006*; *Anzai et al., 2001*; *Baker, 2001*; *Livingstone and Conway, 2007*) of an RF is given by $(R_p - R_q)/(R_p + R_q)$, where $R_p$ is the amplitude at the peak of the 2D Fourier transform of the 2D spatiotemporal RF, found at spatial frequency $F_{\text{space}}$ and temporal frequency $F_{\text{time}}$. $R_q$ is the amplitude at $(F_{\text{space}}, -F_{\text{time}})$ in the 2D Fourier transform. The mean and standard deviations of TDI for experimental data for the cat (*Baker, 2001*) and macaque (*Livingstone and Conway, 2007*) were measured from data extracted from figures in the relevant references (Figure 11A and the low-contrast axis of Figure 3A in these papers respectively).

## Peak temporal frequency

The 2D spatiotemporal RFs were also useful for calculating further temporal response properties of the model. The temporal frequency was calculated as the peak temporal frequency of each spatio-temporal RF as measured from its 2D Fourier transform.

## Code and data availability

All custom code used in this study was implemented in MATLAB and Python. We have uploaded the code to a public Github repository (*Singer, 2018*; copy archived at https://github.com/elifesciences-publications/temporal_prediction_model). The raw auditory experimental data is available at https://osf.io/ayw2p/. The movies and sounds used for training the models are all publicly available at the websites detailed in the Materials and methods.

## Acknowledgements

Nicol Harper was supported by a Sir Henry Wellcome Postdoctoral Fellowship (WT082692) and other Wellcome Trust funding (WT076508AIA, WT108369/Z/2015/Z), by the Department of Physiology, Anatomy and Genetics at the University of Oxford, by Action on Hearing Loss (PA07), and by the Biotechnology and Biological Sciences Research Council (BB/H008608/1). Yosef Singer and Yayoi Teramoto were supported by the Clarendon Fund. Yayoi Teramoto was supported by the Wellcome Trust (10525/Z/14/Z). Andrew King and Ben Willmore were supported by the Wellcome Trust (WT076508AIA, WT108369/Z/2015/Z). We thank Bruno Olshausen for discussions on his model.

## Additional information

### Competing interests

Andrew J King: Senior Editor, *eLife*. The other authors declare that no competing interests exist.

### Funding

| Funder | Grant reference number | Author |
|---|---|---|
| Clarendon Fund | | Yosef Singer<br>Yayoi Teramoto |
| Wellcome | WT10525/Z/14/Z | Yayoi Teramoto |
| Wellcome | WT076508AIA | Ben DB Willmore<br>Andrew J King<br>Nicol S Harper |
| Wellcome | WT108369/Z/2015/Z | Ben DB Willmore<br>Andrew J King<br>Nicol S Harper |
| Wellcome | WT082692 | Nicol S Harper |
| University Of Oxford | | Nicol S Harper |
| Action on Hearing Loss | PA07 | Nicol S Harper |
| Biotechnology and Biological Sciences Research Council | BB/H008608/1 | Nicol S Harper |

The funders had no role in study design, data collection and interpretation, or the decision to submit the work for publication.

### Author contributions

Yosef Singer, Data curation, Software, Formal analysis, Validation, Investigation, Visualization, Methodology, Writing—original draft, Writing—review and editing; Yayoi Teramoto, Software, Formal analysis, Investigation, Methodology, Writing—review and editing; Ben DB Willmore, Data curation, Software, Formal analysis, Supervision, Methodology, Writing—review and editing; Jan WH Schnupp, Resources, Supervision, Funding acquisition, Project administration, Writing—review and

editing; Andrew J King, Supervision, Funding acquisition, Writing—review and editing; Nicol S Harper, Conceptualization, Data curation, Software, Formal analysis, Supervision, Validation, Investigation, Methodology, Writing—original draft, Writing—review and editing, Funding acquisition

## Author ORCIDs
Yosef Singer http://orcid.org/0000-0002-4480-0574
Yayoi Teramoto http://orcid.org/0000-0003-3419-0351
Ben DB Willmore http://orcid.org/0000-0002-2969-7572
Andrew J King https://orcid.org/0000-0001-5180-7179
Nicol S Harper https://orcid.org/0000-0002-7851-4840

## Ethics

Animal experimentation: Auditory RFs of neurons were recorded in the primary auditory cortex (A1) and anterior auditory field (AAF) of 5 pigmented ferrets of both sexes (all > 6 months of age) and used as a basis for comparison with the RFs of model units trained on auditory stimuli. These recordings were performed under license from the UK Home Office and were approved by the University of Oxford Committee on Animal Care and Ethical Review. Full details of the recording methods are described in earlier studies (Willmore et al., 2016; Bizley et al., 2009). Briefly, we induced general anaesthesia with a single intramuscular dose of medetomidine (0.022 mg $\cdot$ kg$^{-1}$ $\cdot$ h$^{-1}$) and ketamine (5 mg $\cdot$ kg$^{-1}$ $\cdot$ h$^{-1}$), which was then maintained with a continuous intravenous infusion of medetomidine and ketamine in saline. Oxygen was supplemented with a ventilator, and we monitored vital signs (body temperature, end-tidal $CO_2$, and the electrocardiogram) throughout the experiment. The temporal muscles were retracted, a head holder was secured to the skull surface, and a craniotomy and a durotomy were made over the auditory cortex. Extracellular recordings were made using silicon probe electrodes (Neuronexus Technologies) and acoustic stimuli were presented via Panasonic RPHV27 earphones, which were coupled to otoscope specula that were inserted into each ear canal, and driven by Tucker-Davis Technologies System III hardware (48 kHz sample rate).

## Decision letter and Author response

Decision letter https://doi.org/10.7554/eLife.31557.034
Author response https://doi.org/10.7554/eLife.31557.035

# Additional files

## Supplementary files
• Transparent reporting form
DOI: https://doi.org/10.7554/eLife.31557.030

## Data availability

All custom code used in this study was implemented in MATLAB and Python. We have uploaded the code to a public Github repository (Singer Y., 2018). The raw auditory experimental data are available at https://osf.io/ayw2p/. The movies and sounds used for training the models are all publicly available at the websites detailed in the Materials and methods.

The following dataset was generated:

| Author(s) | Year | Dataset title | Dataset URL | Database, license, and accessibility information |
| --- | --- | --- | --- | --- |
| Jan Schnupp | 2016 | NetworkReceptiveFields | https://osf.io/ayw2p/ | Available at the Open Science Framework |

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
