## [Decision Letter]

Thank you for submitting your article "Sensory cortex is optimized for prediction of future input" for consideration by *eLife*. Your article has been reviewed by three peer reviewers, and the evaluation has been overseen by Reviewing Editor Jack Gallant and Senior Editor Sabine Kastner. The following individuals involved in review of your submission have agreed to reveal their identity: Rhodri Cusack (Reviewer #1); Laurenz Wiskott (Reviewer #2); Christoph Zetzsche (Reviewer #3).

The reviewers have discussed the reviews with one another, and they and the Reviewing Editor agree that your paper is potentially suitable for publication in *eLife* after appropriate revision. The Reviewing Editor has drafted this decision to help you prepare a revised submission.

The Reviewing Editor hopes that you will address all of the concerns of the authors, most of which are straightforward. But to help you in revision the major issues are listed here:

1) If you look at the reviews you will see that the list of suggestions is very long, but the vast majority of the comments only ask for clarification, they do not require additional data analysis or substantial rewriting. It would be good if you could address all questions in your reply to reviewers and revise the text appropriately where necessary to provide necessary information to the reader.

2) The proposed model is interesting, but it has many components that may differentially contribute to the result, such as the nonlinearity or *L*_1_ regularization. The reviewers felt that the paper would be stronger if the specific effects of these various model component choices were analyzed in a bit more detail, in order to try to pin down more precisely why these components are important. (See for example suggestions of reviewer 2.)

3) In some places the choices made during modelling seemed arbitrary (e.g., the choice of temporal windows and the regularization parameters). Either grid search over a training set should be used to choose these parameters, or hyperparameter modelling should be performed to show that the specific values chosen are not critical, or the choices should be justified. The first of these is obviously most desirable.

4) Some of the figures are difficult to interpret (c.f. Figure 7B and Figure 7—figure supplement 2B). Please try to improve the figures where necessary.

5) Please address the concerns of reviewer 3 regarding the introductory material on temporal prediction and the neurobiological plausibility of the approach.

Reviewer #1:

This manuscript compares two stimulus coding principles that could explain the form of receptive fields in sensory cortex: efficient sparse coding and temporal prediction. A simple temporal prediction model was found create receptive fields that were similar in many ways to those seen in sensory cortex. It performed much better than a sparse coding model.

This manuscript makes an interesting and important contribution. The "sparse coding" hypothesis is popular, both in neuroscience and in artificial intelligence, where autoencoding in deep-neural networks implements efficient coding. The authors argue that for dynamic stimuli, the need to predict what will happen next may be a more important than merely encoding efficiently what has happened recently. Their model is simple and elegant, and the results are convincing.

I felt there were some places where choices were made during modelling that seemed arbitrary – such as the choice of temporal windows and the regularization parameters. The manuscript would be stronger if these choices were either justified, or hyperparameter modelling done to show that the specific values chosen are not critical, to allay concerns readers may have of "p-hacking".

I found the manuscript to be well-structured, thorough and well written, clearly conveying a convincing message.

Reviewer #2:

The paper presents a two-layer network optimized for predicting immediate future sensory input (auditory or visual) from recent past sensory input. The resulting spatio- or spectrotemporal receptive fields, i.e. weight vectors of the first layer, are analyzed and compared with physiological receptive fields. For model comparison, results from a sparse coding network are used.

The results show that the predicting neural network captures receptive field properties fairly well, in particular temporal structure is reproduced much better than by the sparse coding network.

The topic is interesting, and the results are highly relevant to the field. I must add, however, that I have not followed the field recently, so I cannot really tell, whether some similar work has been published recently. But the authors seem to have done a careful literature research and discuss alternative approaches fairly.

The paper is well structured and has obviously been written very carefully. I have rarely reviewed a manuscript that feels so ready for publication. So, I am tempted to recommend the paper for publication as it is.

There is just one issue I would invite the authors to consider a bit further: The claim of the paper is that the objective of temporal prediction results in the receptive field properties found. But there are additional factors, such as the nonlinearity and the *L*_1_ regularization, that contribute to it. The authors have investigated this to some extent. For example, they find that receptive fields are seriously degraded if the nonlinearity is replaced by a linear activation function. My suggestion is to try to pin down, what objective is implicitly added by the nonlinearity and the *L*_1_ regularization. I suggest to perform a similar experiment as in Figure 8, but with a sparseness or independence measure rather than final validation loss. This could also be done on the hidden units.

I suggest this, because I believe that temporal prediction alone does not do the trick. I feel it must be combined with some sparseness or independence objective to yield the receptive fields. And I feel that this missing objective is implicitly added by the nonlinearity and the *L*_1_ regularization. Making this more transparent would be great and the suggested experiment should be very easy to do.

Reviewer #3:

The authors propose a new principle for the development of cortical receptive fields which combines the concepts of predictive coding and sparse coding. They train a three-layer network with one hidden layer in order to predict the future visual spatial input or the future auditory auditory spectro-temporal input from the recent spatio-/spectro-temporal input, subject to a sparsity constraint.

They perform this training for two examples: for an auditory network, based on training data which contain human speech, animal vocalization and inanimate natural sounds, and for a visual network based on training data with movies of wildlife in natural settings.

They compare the resulting networks to real cortical neurons from A1 and V1 and to an alternative sparse coding approach intended to provide a sparse representation of the complete spatio/spectro-temporal input.

For their comparison they consider the spectrotemporal and spatiotemporal receptive fields and various population measures, e.g. the temporal decay of power in the receptive fields, the temporal span of excitation an inhibition, orientation and frequency tuning properties, and receptive field dimensions.

Except for orientation, for which the majority of visual units is restricted in their orientation preference to horizontal and vertical orientations, the proposed model can capture neural tuning properties as well as the established models. And in case of the asymmetric emphasis of the most recent past it can even provide a better description.

In my opinion this is a quite interesting paper. First, it presents a novel approach which unifies the principles of sparse coding and of temporal prediction. This combination enables the explanation of a large set of spatio-/spectro-temporal tuning properties within one single integrated framework. Second, the authors have an important point in stressing the asymmetry of the temporal response with its emphasis of the most recent past, as observed in typical cortical neurons. This is indeed a property that other learning schemes, like sparse coding, by the very nature of their objective functions, cannot produce.

There are some points that, in my view, need to be clarified or described in more detail. In the following I describe the modifications and additions which I assume to be helpful in a revision of the paper. Due to my background I will put more emphasis on the visual aspects.

1) The description of the history of the concept of temporal prediction is not clear enough, both in the introduction and in the discussion. I am aware of the pressure for novelty in current science but in my view, there is sufficient novelty in the suggested model to allow the authors to avoid such ambiguities. Currently, the paper might be misinterpreted by a swift non-specialist reader as if the concept of the "prediction of the immediate future" is a novel principle being introduced here (Introduction; Discussion section: "We hypothesized"). Only a few selected papers are cited directly in this context (only Bialek in the Introduction), and in the Discussion section they are characterized as unspecific: "The temporal prediction principle we describe.… has been described in a very general manner"(reference only to Bialek, Palmer). Other references exist but are spread out through the further text. But of course, the principle as such has a long history, there are numerous papers which describe the prediction of the future sensory input as an important goal of neural information processing. I am no specialist, and this is not comprehensive but early examples are corollary discharge theories, and already Sutton and Barto, (1981) and Srinivasan et al., (1982), for example, considered the temporal dimension of prediction. Motion extrapolation has also been interpreted as prediction computed in visual cortex (Nijhawan, 1994). A further, canonical example of a method for the optimal prediction of the future sensory input is the Kalman filter, as considered by, e.g., Rao, (1999). I suggest that the authors devote one paragraph to the history of the concept, with all the appropriate references included there, and then make precisely clear in which aspects their novel contribution extends beyond these earlier approaches.

2) Neurobiological plausibility of the approach: I do not think that the authors have to be as clear about the neural implementation of the suggested architecture as the other predictive coding approaches, but at least some rough or speculative ideas should be presented: What is the status of the second-order units? Where in the cortex are they (V2?) and what do they encode? Really the future INPUT itself? That is, they have no selectivity, no tuning properties? Have such units been observed? Where and how is the prediction error computed? Does this model not require a bypass line which brings the retinal spatial input directly to V1 or V2 to enable the comparison?.…

3) The sparse coding data appear quite unusual. Why are the units not more "localized" in the temporal dimension, in particular for the visual model? Furthermore, it seems as if the visual sparse model is not used in the usual overcomplete regime. I also would have expected a more concentrated distribution of the temporal span of excitation and inhibition for a typical sparse coding model. Please discuss this in the paper.

4) Subsection “Model receptive fields” by inspection: is there really no other possibility to determine the optimal hyperparameters of the sparse coding model? A fit to the neural data? You have Figure 8—figure supplement 1B anyway. Why have you not made use of it? One could use only a training subset, if this seems critical issue. And one can include KS measures of other tuning properties.

5) Subsection “Addition of Gaussian noise”: Noise. For me the use of noise in this investigation is somewhat unclear. First, it seems to favor the prediction model over the sparse model, which is more susceptible to noise. Second, the noise level used appears unusually strong (is this dB?). This issue should be clearly motivated and discussed in the main text of the paper.

6) Subsection “Model receptive fields”: I am a bit skeptic with respect to the sign-flipping of excitation and inhibition. The argument that the signs could as well be flipped if this is done for the first-order and the second-order units alike appears only valid because any specification and relation to real neurons is omitted for the second-order units. In fact, this sign-flipping will inevitably imply a prediction of how excitation and inhibition operate in the second-order units.

Furthermore, if this argumentation would be accepted then one could arbitrarily flip signs to the desired result in any learning model, because one can always argue that appropriate sign flips at some subsequent processing stages could compensate for this. Used in this way, excitation and inhibition would lose any meaning.

It is perfectly ok for me if a model is agnostic with respect to the correct prediction of excitation and inhibition, a model does not have to be perfect in all aspects. But if this is the case this should be clearly visible for the reader in the presented receptive field plots. (This does not exclude to use of an appropriate sign-flip in population measures.)

7) Figure 4—figure supplement 2 and Figure 4—figure supplement 3: Two separate populations? Visual inspection of these figures suggests the possible existence of two distinct populations. Is this related to separability, or blob-like units, or both? (Is ordering according to separability?) I am not sure about the current state in the field, but I remember a discussion about the existence of two distinct populations as opposed to a continuous distribution for separability. This issue should be described and discussed.

8) ibid. The percentage of blob-like units appears quite high. Is this percentage comparable to the neural data? Or only if the mouse data, which are special in this respect, are being included?

9) Figure 4—figure supplement 6 The percentage of blob-like units seems to be substantially reduced in comparison to the noisy case (Figure 4—figure supplement 2). Is there a systematical relation between high noise levels and the emergence of blob-like units? Please discuss this issue in the paper.

10) ibid. It is difficult to understand the relation between Figure 4—figure supplement 2 and Figure 7C as opposed to Figure 4—figure supplement 6 and Figure 7—figure supplement 2F. Can you describe how properties of the receptive field plots relate to properties of the population distribution in these two cases?

11) Figure 7 and others Spatiotemporal population properties: Although the article is about spatiotemporal processing it provides only two population measures of purely static spatial properties and one of a purely temporal property for the vision case. Spatiotemporal measures of particular interest would be: DSI (directional selectivity index)/TDI (tilt direction index) (direction selectivity is considered to be a major spatiotemporal property of visual cortex); if possible: a scatter plot of temporal frequency vs. spatial frequency; population distribution of motion tuning. The necessary data for these plots should be already available.

12) Figure 7B and Figure 7—figure supplement 2B: The orientation scatter plot is visually difficult to interpret. The quantitative degree of concentration of the preferred orientations on the vertical and horizontal orientations as opposed to the oblique orientations remains unclear. Please provide either an orientation histogram or the percentage of units which fall into the 30-60, 120-150 deg range. In both cases the lowest spatial frequencies should be omitted for the analysis.

13) Figure 7—figure supplement 1: I am a bit surprised that only 289/400 sparse-coding units can be fitted by a Gabor function. Why is this? Usually most sparse coding units have a good Gabor fit. And with such a high percentage excluded I see the risk of a systematic bias regarding certain tuning parameters.

14) Figure 7C and Figure 7—figure supplement 2F seem to indicate that the model produces two distinct sub-populations with respect to receptive field parameters n_x and n_y. It should be discussed whether this is the case, and if yes, whether it is systematically related to other parameters (selectivities) of the units. Could this be related related to the two apparent sub-populations regarding separability, blob-like shapes, cf. Figure 4—figure supplement 2? And has such a tendency has also been observed in neural data? In contrast, Ringach, (2004) claimed clustering around a one-dimensional curve. Please describe and discuss in text.

15) Figure 8 and subsection “Implementation details: Are we expected to see here that 1600 hidden units are a distinguished optimum? Does this figure not tell us that the prediction error does not substantially depend on the number? And when a biological system could achieve basically the same prediction quality with 100 neurons why should it then invest 1500 additional units for such a small advantage?

16) Figure 8 and Figure 8—figure supplement 1. The comparison between the models prediction capability and the similarity of the model units to real units should not only be presented for the auditory neurons but also for the visual neurons (according to the text the data seem to be already available).

17) Subsection “Optimising predictive capacity” It is not clear whether the similarity measure (only the span of temporal and frequency tuning is considered) is fair or biased for the comparison with the sparse coding model. What will happen, for example, if the distribution of orientation preference would be used instead (for the visual model)? Would then the sparse coding model appear more similar to the real neurons than the prediction model?

Please motivate and discuss.

18) Discussion section: I cannot follow the argumentation that the class of prediction models (or this specific model) should somehow be unique with respect to the ability to provide an independent criterion for the selection of hyperparameters. Should this somehow be a principle, or a logical conclusion? Or an empirical observation only for this special case? Is it logically impossible that we find a measure, for example something entropy-related or whatever, for sparse coding that could have the same status? Please clarify.

19) Subsection “Visual normative models”:does not refer to prediction of the future: Why not? Is not the temporal prediction made in these models that the future spatial pattern is the same as the previous spatial pattern?

20) Subsection “Visual normative models”:The model is selective, throws away information, as opposed to the information preserving properties of other models. But is this really a good strategy for *early* stages of a multi-stage information processing system? Does the principle of least commitment not suggest just the opposite strategy?

21) For the Discussion section it would be of interest to consider the contributions from the two components of the prediction model, i.e., which properties of the units are genuinely caused by prediction and which are more due to the sparse coding part?

[Editors' note: further revisions were requested prior to acceptance, as described below.]

Thank you for resubmitting your work entitled "Sensory cortex is optimized for prediction of future input" for further consideration at *eLife*. Your revised article has been favorably evaluated by Sabine Kastner (Senior Editor/Reviewing Editor), and two reviewers.

The manuscript has been improved but there are some remaining issues that need to be addressed before acceptance, as outlined below:

The authors have addressed my comments in a careful manner. In particular, I appreciate that they made considerable and appropriate changes to text and figures instead of just providing arguments in the response letter. From my view, the manuscript is basically ready for publication. I have a few final minor suggestions.

1).… We examined linear aspects of the tuning of the output units for the visual temporal prediction model using a response-weighted average to white noise input and found punctate un-oriented RFs that decay into the past..…

This is interesting. Can you mention this somewhere in the text?

2) I understand that the model, by its very nature would not care about the sign. But the fact remains that you have an output of a model and you post hoc manipulate this output to obtain a "better suited" presentation (e.g., to ease comparison). My only point is that it should be totally clear to even a superficial reader that such a post hoc change has been applied. So please just include an appropriate sentence that makes this clear, e.g.:

Note that the model does not care about the sign (excitation/inhibition) and thus provides no systematic prediction of it. We hence switched the signs of the respective receptive fields of the model output appropriately to obtain receptive fields which all have positive leading excitation.

(3) Can you mention this alternative goal of least commitment somewhere in the discussion? And the empirical question.

---

## [Author Response]

The reviewers have discussed the reviews with one another, and they and the Reviewing Editor agree that your paper is potentially suitable for publication in eLife after appropriate revision. The Reviewing Editor has drafted this decision to help you prepare a revised submission.The Reviewing Editor hopes that you will address all of the concerns of the authors, most of which are straightforward. But to help you in revision the major issues are listed here:1) If you look at the reviews you will see that the list of suggestions is very long, but the vast majority of the comments only ask for clarification, they do not require additional data analysis or substantial rewriting. It would be good if you could address all questions in your reply to reviewers and revise the text appropriately where necessary to provide necessary information to the reader.

We hope that our explanations in this document and clarifications throughout the main text of the paper address these points. We have addressed all of the comments in this document and made corresponding changes in the text.

2) The proposed model is interesting, but it has many components that may differentially contribute to the result, such as the nonlinearity or L_1_ regularization. The reviewers felt that the paper would be stronger if the specific effects of these various model component choices were analyzed in a bit more detail, in order to try to pin down more precisely why these components are important. (See for example suggestions of reviewer 2.)

In preparing the original manuscript, we performed a grid search over the *L*_1_ regularization parameter and number of hidden units. The effect of changing these parameters on the predictive capacity of the model are shown for the auditory model in Figure 8. We did not previously show the effects of these parameters on the receptive field structure. We have now produced interactive figures which illustrate this for visual and auditory RFs (Figure 8—figure supplement 2 and Figure 8—figure supplement 3; https://yossing.github.io/temporal_prediction_model/figures/interactive_supplementary_figures.html). The effects of these parameters are now described in the main text (see paragraph below).

We have also explored the effects of the activation function (nonlinearity). We implemented versions of the network with tanh and rectified linear activation functions and found that the choice of nonlinearity does not have a decisive effect on RF structure However, if linear activation is used its RFs do not look like those seen in V1 or A1, as discussed in the main text (see paragraph below).

We have also explored other components of the model (see the response to point 3). However, two particular components of the model appear to be particularly important for prediction and having RFs that match the biology; having a non-linearity and having the correct amount of *L*_1_ weight regularization. We suspect that there are two reasons why having appropriate *L*_1_ regularization is likely to be important; first to avoid overfitting and hence find the most predictive code, and second to mimic the efficiency constraints on connectivity of the nervous system due to space and energy limitations. We suspect that a reason why having a nonlinearity is likely to be important, is that the future input depends non-linearly on the past input.

Reviewer #2 makes the interesting point that although the sparseness or independence of the hidden unit activities is not an explicit goal of the model, this may emerge implicitly in cases where the model’s RFs are most similar to the neural data (and where prediction error is lowest). To test for this, we measured the sparsity of the trained model’s hidden unit activities (by the measure of Vinje and Gallant, (2000)) in response to the natural input validation set. Examining the relationship between predictive capacity and sparsity, over a range of *L*_1_ weight regularization strength and hidden units, we do not find a clear monotonic relationship. Indeed, the hidden unit and *L*_1_ regularization combination with the best prediction was not the sparsest model, but of intermediate sparsity over the span we explored.

Addressing these points and others, we have now added a new section to the Results section titled “Variants of the temporal prediction model”. The relevant part at the start of this new section reads:

“The change in the qualitative structure of the RFs as a function of the number of hidden units and *L*_1_ regularization strength, for both the visual and auditory models, can be seen in the interactive supplementary figures (Figure 8—figure supplements 2-3; https://yossing.github.io/temporal_prediction_model/figures/interactive_supplementary_figures.html) [...] The RFs also did not change form or polarity over time, but simply decayed into the past.”.

3) In some places the choices made during modelling seemed arbitrary (e.g., the choice of temporal windows and the regularization parameters). Either grid search over a training set should be used to choose these parameters, or hyperparameter modelling should be performed to show that the specific values chosen are not critical, or the choices should be justified. The first of these is obviously most desirable.

We have explored the effects of all of the crucial parameters, and we explain these results in the new section “Variants on the temporal prediction model”.

Regularization parameter and hidden unit number: In all the model results shown in the original manuscript, we did not choose the values of these parameters, but used the parameters which provided the optimal prediction within a grid search (see response to point 2, above).

Temporal window into the past: We have explored the effect of different temporal windows. As can be seen in Figure 6B and Figure 7A, most of the energy of the real and model RFs is within 100ms (20 time steps) in the auditory case and within 160ms (4 time steps) in the visual case. Hence, the temporal window into the past was chosen to be slightly larger than these values, and so long as the window into the past is sufficiently long, it’s length is not a critical.

Temporal window into the future: We have explored the effect of different temporal windows into the future. We found that increasing the number of time steps being predicted had little effect on the RFs of the auditory model either qualitatively or by the KS similarity measure. In the visual case, it caused the RFs to be more restricted in space and increased the proportion of blob-like units.

Noise: In the Supplementary Material of the original manuscript we included figures showing that the effects of the input noise were subtle and not-critical to the form of RF that we see. We now discuss this in the main Results section as requested by reviewer #3.

Nonlinearity: Although a non-linearity is critical to achieve our results, with the RFs appearing quite different and less like those seen in V1 or A1 when no non-linearity is used, the exact choice of non-linearity (sigmoid, tanh, or rectified linear) was not critical, and the RFs seen were similar (see our response to point 2 above).

In addition to the parts of the new section given in response to the Editor’s previous point, the relevant part of this new section subsection “Visual normative models”) reads:

“The temporal prediction model and sparse coding model results shown in the main figures of this paper were trained on inputs with added Gaussian noise (6dB SNR), mimicking inherent noise in the nervous system. To determine the effect of adding this noise, all models were also trained without noise, producing similar results (Figure 4—figure supplements 5–7; Figure 5—figure supplements 3–5; Figure 6—figure supplement 1; Figure 7—figure supplements 2-3). The results were also robust to changes in the duration of the temporal window being predicted. We trained the auditory model to predict a span of either 1, 3, 6, or 9 time steps into the future and the visual model to predict 1, 3 or 6 time steps into the future. For the auditory case, we found that increasing the number of time steps being predicted had little effect on the RF structure, both qualitatively and by the KS measure of similarity to the real data. In the visual case, Gabor-like units were present in all cases. Increasing the number of time steps made the RFs more restricted in space and increased the proportion of blob-like RFs.”

4) Some of the figures are difficult to interpret (c.f. Figure 7B and Figure 7—figure supplement 2B). Please try to improve the figures where necessary.

We have changed these figures, as suggested by the reviewers, in order to make them more interpretable. We have added an additional panel to Figure 7 and corresponding supplementary figures showing a histogram of orientation tuning preferences for the model units. We have also added insets to Figure 6 and corresponding supplementary figures.

5) Please address the concerns of reviewer 3 regarding the introductory material on temporal prediction and the neurobiological plausibility of the approach.

We have now added the following paragraph to the Introduction on predictive normative models of sensory processing.

“The idea that prediction is an important component of perception dates at least as far back as Helmholtz^18,19^, although what is meant by prediction and the purpose it serves is quite varied between models incorporating it^20,21^. […] Our model relates to the predictive information approach in that it is optimized to predict the future from the past, but it has a combination of characteristics, such a non-linear encoder and sparse weight regularization, which have not previously been explored for such an approach.”

We have also extended the paragraph exploring the neurobiological plausibility of the model (subsection “Strengths and limitations of the temporal prediction model”). It now reads:

“Finally, it is interesting to consider possible more explicit biological bases for our model. We envisage the input units of the model as thalamic input, and the hidden units as primary cortical neurons. […] Finally, it is important to note that, although the biological plausibility of backpropagation has long been questioned, recent progress has been made in developing trainable networks that perform similarly to artificial neural networks trained with backpropagation, but with more biologically plausible characteristics^77^, for example, by having spikes or avoiding the weight transport problem^78^”

Reviewer #1:This manuscript compares two stimulus coding principles that could explain the form of receptive fields in sensory cortex: efficient sparse coding and temporal prediction. A simple temporal prediction model was found create receptive fields that were similar in many ways to those seen in sensory cortex. It performed much better than a sparse coding model.This manuscript makes an interesting and important contribution. The "sparse coding" hypothesis is popular, both in neuroscience and in artificial intelligence, where autoencoding in deep-neural networks implements efficient coding. The authors argue that for dynamic stimuli, the need to predict what will happen next may be a more important than merely encoding efficiently what has happened recently. Their model is simple and elegant, and the results are convincing.

Thank you.

I felt there were some places where choices were made during modelling that seemed arbitrary – such as the choice of temporal windows and the regularization parameters. The manuscript would be stronger if these choices were either justified, or hyperparameter modelling done to show that the specific values chosen are not critical, to allay concerns readers may have of "p-hacking".

In order to address this point, we have performed a grid search over a large space of hyperparameter values. The results of this search are outlined above in response to the editor’s third point. From these plots, and the associated text, one can see that there is substantial robustness to the exact modelling choices.

The duration of the temporal window into the past is relatively unimportant because most of the energy is in the most recent few steps (Figure 6B and Figure 7A), and so long as it is long enough to capture this span it suffices. We have now explored the effect of the duration of the temporal window into the future, and as we mention in our reply to point 3 of the editor, this has little effect on the RFs. The regularization parameter was chosen by selecting the value that best predicts the future of a held-out validation set (see Figure 8 and the corresponding subsection “Optimizing Predictive Capacity”.

I found the manuscript to be well-structured, thorough and well written, clearly conveying a convincing message.

Thank you.

Reviewer #2:The paper presents a two-layer network optimized for predicting immediate future sensory input (auditory or visual) from recent past sensory input. The resulting spatio- or spectrotemporal receptive fields, i.e. weight vectors of the first layer, are analyzed and compared with physiological receptive fields. For model comparison, results from a sparse coding network are used.The results show that the predicting neural network captures receptive field properties fairly well, in particular temporal structure is reproduced much better than by the sparse coding network.The topic is interesting, and the results are highly relevant to the field. I must add, however, that I have not followed the field recently, so I cannot really tell, whether some similar work has been published recently. But the authors seem to have done a careful literature research and discuss alternative approaches fairly.The paper is well structured and has obviously been written very carefully. I have rarely reviewed a manuscript that feels so ready for publication. So, I am tempted to recommend the paper for publication as it is.

We appreciate the reviewer’s very positive comments.

There is just one issue I would invite the authors to consider a bit further: The claim of the paper is that the objective of temporal prediction results in the receptive field properties found. But there are additional factors, such as the nonlinearity and the L_1_ regularization, that contribute to it. The authors have investigated this to some extent. For example, they find that receptive fields are seriously degraded if the nonlinearity is replaced by a linear activation function. My suggestion is to try to pin down, what objective is implicitly added by the nonlinearity and the L_1_ regularization. I suggest to perform a similar experiment as in Figure 8, but with a sparseness or independence measure rather than final validation loss. This could also be done on the hidden units.I suggest this, because I believe that temporal prediction alone does not do the trick. I feel it must be combined with some sparseness or independence objective to yield the receptive fields. And I feel that this missing objective is implicitly added by the nonlinearity and the L_1_ regularization. Making this more transparent would be great and the suggested experiment should be very easy to do.

We have now examined the effects of these choices in some detail -- see our response to the second point of the Editor.

Reviewer #3:The authors propose a new principle for the development of cortical receptive fields which combines the concepts of predictive coding and sparse coding. They train a three-layer network with one hidden layer in order to predict the future visual spatial input or the future auditory auditory spectro-temporal input from the recent spatio-/spectro-temporal input, subject to a sparsity constraint.They perform this training for two examples: for an auditory network, based on training data which contain human speech, animal vocalization and inanimate natural sounds, and for a visual network based on training data with movies of wildlife in natural settings.They compare the resulting networks to real cortical neurons from A1 and V1 and to an alternative sparse coding approach intended to provide a sparse representation of the complete spatio/spectro-temporal input.For their comparison they consider the spectrotemporal and spatiotemporal receptive fields and various population measures, e.g. the temporal decay of power in the receptive fields, the temporal span of excitation an inhibition, orientation and frequency tuning properties, and receptive field dimensions.Except for orientation, for which the majority of visual units is restricted in their orientation preference to horizontal and vertical orientations, the proposed model can capture neural tuning properties as well as the established models. And in case of the asymmetric emphasis of the most recent past it can even provide a better description.In my opinion this is a quite interesting paper. First, it presents a novel approach which unifies the principles of sparse coding and of temporal prediction. This combination enables the explanation of a large set of spatio-/spectro-temporal tuning properties within one single integrated framework. Second, the authors have an important point in stressing the asymmetry of the temporal response with its emphasis of the most recent past, as observed in typical cortical neurons. This is indeed a property that other learning schemes, like sparse coding, by the very nature of their objective functions, cannot produce.

Thank you.

There are some points that, in my view, need to be clarified or described in more detail. In the following I describe the modifications and additions which I assume to be helpful in a revision of the paper. Due to my background I will put more emphasis on the visual aspects.1) The description of the history of the concept of temporal prediction is not clear enough, both in the introduction and in the discussion. I am aware of the pressure for novelty in current science but in my view, there is sufficient novelty in the suggested model to allow the authors to avoid such ambiguities. Currently, the paper might be misinterpreted by a swift non-specialist reader as if the concept of the "prediction of the immediate future" is a novel principle being introduced here (Introduction; Discussion section: "We hypothesized"). Only a few selected papers are cited directly in this context (only Bialek in the Introduction), and in the Discussion section they are characterized as unspecific: "The temporal prediction principle we describe.… has been described in a very general manner"(reference only to Bialek, Palmer). Other references exist but are spread out through the further text. But of course, the principle as such has a long history, there are numerous papers which describe the prediction of the future sensory input as an important goal of neural information processing. I am no specialist, and this is not comprehensive but early examples are corollary discharge theories, and already Sutton and Barto, (1981) and Srinivasan et al., (1982), for example, considered the temporal dimension of prediction. Motion extrapolation has also been interpreted as prediction computed in visual cortex (Nijhawan, 1994). A further, canonical example of a method for the optimal prediction of the future sensory input is the Kalman filter, as considered by, e.g., Rao, (1999). I suggest that the authors devote one paragraph to the history of the concept, with all the appropriate references included there, and then make precisely clear in which aspects their novel contribution extends beyond these earlier approaches.

We have added a new paragraph to the Introduction to address this. This paragraph points out the novel aspects of our model’s structure. See our response to the fifth point of the Editor. We have also expanded the final paragraph of the Introduction to set out a central novel contribution of our model, which is that it successfully explains temporal properties of both V1 and A1 neurons, something previous models have not been able to do. The expanded part of the final paragraph (Introduction) reads: “Here we show using qualitative and quantitative comparisons that, for both V1 and A1 RFs, these shortcomings are largely overcome by the temporal prediction approach. This stands in contrast to previous models”

2) Neurobiological plausibility of the approach: I do not think that the authors have to be as clear about the neural implementation of the suggested architecture as the other predictive coding approaches, but at least some rough or speculative ideas should be presented: What is the status of the second-order units? Where in the cortex are they (V2?) and what do they encode? Really the future INPUT itself? That is, they have no selectivity, no tuning properties? Have such units been observed? Where and how is the prediction error computed? Does this model not require a bypass line which brings the retinal spatial input directly to V1 or V2 to enable the comparison?.…

We have now addressed this more fully in the paper -- see our response to the fifth point of the Editor. We now describe how the output units of our model would represent the prediction of the input or the prediction error and note that signals relating to prediction error have been found in A1 (Rubin et al., 2016). These properties of the output units are non-linear and wouldn’t be fully captured by linear RF methods. We examined linear aspects of the tuning of the output units for the visual temporal prediction model using a response-weighted average to white noise input and found punctate un-oriented RFs that decay into the past. We suspect that no bypass line would be required, just appropriate temporal asymmetries in input and/or synaptic plasticity mechanisms.

3) The sparse coding data appear quite unusual. Why are the units not more "localized" in the temporal dimension, in particular for the visual model?

For the auditory data, our results are not unusual, see Carlson et al., (2012) and Carlin and Elhilali, (2013), where RFs that fully span the temporal dimension are common.

For the visual data, our results are also not unusual. If you examine the figure of van Hateran and Ruderman, (1998), they only show four units. It is not clear whether the units they show are reflective of the tuning properties of the entire population or have been selected because they are temporally localised. In the sparse coding results we presented, although we also see some units which are temporally localised, the majority were not. We have now run the sparse coding model in the overcomplete regime as is commonly done in the literature, and while the majority of spatiotemporal RFs are still not temporally localized, there are more examples that are. Notably, in a later paper looking at ICA of spatiotemporal visual inputs by Hyvärinen et al. (2003), which does show the full set of RFs, there are very few examples of units whose RFs are temporally localized, while the vast majority are not.

Furthermore, it seems as if the visual sparse model is not used in the usual overcomplete regime.

We have now run the sparse coding model with more hidden units and restricted the results presented in both the visual and auditory cases to configurations where the sparse coding models were run in the overcomplete regime. We have added a sentence to clarify this in the Materials and methods section. It reads:

“In both cases, the model configurations chosen were restricted to those trained in an overcomplete condition (having more units than the number of input variables) in order to remain consistent with previous instantiations of this model^4,5,11^…we selected a sparse coding network with 1600 units…in the auditory case (Figure 5 and Figure 6).In the visual case, the network selected was trained with 3200 units, λ=10^0.5^, learning rate = 0.05 and 100 mini-batches.”

I also would have expected a more concentrated distribution of the temporal span of excitation and inhibition for a typical sparse coding model. Please discuss this in the paper.

Our results are in keeping with previous studies in this regard (see Carlson et al., (2012) and Carlin and Elhilali, (2013)). We have added a sentence to the Results section highlighting this point. It reads: “The sparse coding model shows a wide range of temporal spans of excitation and inhibition, in keeping with previous studies^11,48^.”

4) Subsection “Model receptive fields” by inspection: is there really no other possibility to determine the optimal hyperparameters of the sparse coding model? A fit to the neural data? You have Figure 8—figure supplement 1B anyway. Why have you not made use of it? One could use only a training subset, if this seems critical issue. And one can include KS measures of other tuning properties.

This is a good point. We simply chose the ones that, by eye, presented the sparse coding model in the best light. However, if we choose the model in which RFs lie at the minimum by the KS measure in the auditory case, they do not look much different from the ones we chose. We have now changed the set of sparse coding model units in the auditory case to those that are most similar to the real neurons according to the KS measure, while still being overcomplete. In the visual case, as we do not perform a KS measure due to the limited amount of data, particularly pertaining to temporal response properties (see response to point sixteen below), this model was chosen by inspection. We have changed the text in subsection “Sparse coding model” to reflect this. It now reads: “…we chose the network that produced basis functions whose receptive fields were most similar to those of real neurons. In the auditory case, this was determined using the mean KS measure of similarity (Figure 8—figure supplement 1). In the visual case, as a similarity measure was not performed, this was done by inspection.”

5) Subsection “Addition of Gaussian noise”: Noise. For me the use of noise in this investigation is somewhat unclear. First, it seems to favor the prediction model over the sparse model, which is more susceptible to noise. Second, the noise level used appears unusually strong (is this dB?). This issue should be clearly motivated and discussed in the main text of the paper.

The temporal prediction and sparse coding models were also run without noise, and the results in all cases were similar. We have now added a paragraph in the new Results section which motivates and discusses the influence of the noise in the model as suggested by the reviewer. The paragraph in subsection “Variants of the temporal prediction model” reads: “The temporal prediction model and sparse coding model results shown in the main figures of this paper were trained on inputs with added Gaussian noise (6dB SNR), mimicking inherent noise in the nervous system. To determine the effect of adding this noise, all models were also trained without noise, producing similar results (Figure 4—figure supplements 5-7; Figure 5—figure supplements 3–5; Figure 6—figure supplement 1; Figure 7—figure supplements 2–3).”

We have also added clarifying text to the caption of Figure 7—figure supplement 2 highlighting the main effect of the addition of noise to the quantitative results in the visual case. It reads: “The addition of noise only leads to subtle changes in the RFs; most apparently, there are more units with RFs comprising multiple short subfields (forming an increased number of points towards the lower right quadrant of g) than is seen in the case when noise is used.”

We have also added text to the caption of Figure 6—figure supplement 1 highlighting the main effects of the addition of noise to the model in the auditory case. It reads: “The addition of noise leads to subtle changes in the RFs. Without noise, the inhibition in the temporal prediction model tends to be slightly less extended and the RFs a little less smooth (see Figure 4, Figure 4—figure supplement 5 for qualitative comparison).”

The addition of noise does not seem to negatively impact the sparse coding model’s results. For the sparse coding model in the visual case, the results are much the same with noise as without (Figure 5—figure supplement 1, Figure 5—figure supplement 2, Figure 5—figure supplement 4Figure 5—figure supplement 5 and Figure 7—figure supplement 2 and Figure 7—figure supplement 3). In the auditory case without added noise, the RFs tended to have somewhat smoother backgrounds, but were otherwise much the same in form as when noise was added (Figure 5, Figure 5—figure supplement 3). Quantitative comparison of the models trained on auditory inputs without added noise shows only subtle differences from the case with noise (Figure 6, Figure 6—figure supplement 1).

The signal-to-noise ratio (SNR) given was the variance of the signal divided by the variance of the noise, not in decibels. Apologies this was not more clear. We have now give the SNR in decibels, a more conventional measure. In decibels, the SNR is 6dB. Hence the noise is weaker than may have been implied.

6) Subsection “Model receptive fields”: I am a bit skeptic with respect to the sign-flipping of excitation and inhibition. The argument that the signs could as well be flipped if this is done for the first-order and the second-order units alike appears only valid because any specification and relation to real neurons is omitted for the second-order units. In fact, this sign-flipping will inevitably imply a prediction of how excitation and inhibition operate in the second-order units.Furthermore, if this argumentation would be accepted then one could arbitrarily flip signs to the desired result in any learning model, because one can always argue that appropriate sign flips at some subsequent processing stages could compensate for this. Used in this way, excitation and inhibition would lose any meaning.It is perfectly ok for me if a model is agnostic with respect to the correct prediction of excitation and inhibition, a model does not have to be perfect in all aspects. But if this is the case this should be clearly visible for the reader in the presented receptive field plots. (This does not exclude to use of an appropriate sign-flip in population measures.)

We are agnostic as to what the biological analogue of the output units is (see reply to the Editor’s fifth point). Therefore, the only units for which we make a direct comparison with data are the hidden units.

The sign flipping makes no difference to the function of the network, as we describe in subsection “Model receptive fields”. This degeneracy arises because, in the network, the units’ activation functions are symmetric, whereas for real neurons, high firing rates are meaningfully different from low ones. We therefore need to choose between two equivalent descriptions of the receptive fields. We agree that flipping the signs of each pixel in each RF or of each RF independently to maximize similarity to the data would be arbitrary and unjustified. However, this is not what we do. What we do instead is make all the units have positive leading excitation, which reflects the structure of the majority of cortical units and allows the reader to make a visual comparison to the data.

7) Figure 4—figure supplement 2 and Figure 4—figure supplement 3: Two separate populations? Visual inspection of these figures suggests the possible existence of two distinct populations. Is this related to separability, or blob-like units, or both? (Is ordering according to separability?) I am not sure about the current state in the field, but I remember a discussion about the existence of two distinct populations as opposed to a continuous distribution for separability. This issue should be described and discussed.

We discovered that some of the movie snippets we were using to train the model contained writing at a very high contrast. These examples have since been removed and the results updated with RFs obtained when the model was trained on inputs without any of these writing snippets included. In the updated results, the subpopulation of small blob-like units specified by the reviewer is no longer present (Figure 4—Figure supplement 2 and Figure 4—figure supplement 3).

8) ibid. The percentage of blob-like units appears quite high. Is this percentage comparable to the neural data? Or only if the mouse data, which are special in this respect, are being included?

See above – this subpopulation of small, blob-like units is no longer present in the results presented (Figure 4—Figure supplement 2 and Figure 4—figure supplement 3).

9) Figure 4—figure supplement 6 The percentage of blob-like units seems to be substantially reduced in comparison to the noisy case (Figure 4—figure supplement 2). Is there a systematical relation between high noise levels and the emergence of blob-like units? Please discuss this issue in the paper.

As mentioned in response to the previous two points, we discovered and removed visual training examples that contained writing. In the updated results, the subpopulation of small blob-like units that differentiated the results in the noise and non-noise cases is no longer present and the results are now more similar between the two cases.

10) ibid. It is difficult to understand the relation between Figure 4 supplement 2 and Figure 7c as opposed to figure 4—figure supplement 6 and Figure 7—figure supplement 2F. Can you describe how properties of the receptive field plots relate to properties of the population distribution in these two cases?

We now discuss these distributions in more detail. In subsection “Quantitative analysis of visual results”, we write: “The distribution of units extends along a curve from blob-like RFs, which lie close to the origin in this plot, to stretched RFs with several subfields, which lie further from the origin.” In the caption of Figure 7—figure supplement 2, we write “The addition of noise in both cases only leads to subtle changes in the RFs; most apparently, there are more units with RFs comprising multiple short subfields (forming an increased number of points towards the lower right quadrant of g) than is seen in the case when noise is used.”

11) Figure 7 and others Spatiotemporal population properties: Although the article is about spatiotemporal processing it provides only two population measures of purely static spatial properties and one of a purely temporal property for the vision case. Spatiotemporal measures of particular interest would be: DSI (directional selectivity index)/TDI (tilt direction index) (direction selectivity is considered to be a major spatiotemporal property of visual cortex); if possible: a scatter plot of temporal frequency vs. spatial frequency; population distribution of motion tuning. The necessary data for these plots should be already available.

We have now measured the TDI of the model population and added a section specifying the result in the main text (subsection “Quantitative analysis of visual results”). It reads: “In addition to this, we measured the tilt direction index (TDI) of the model units from their 2D spatiotemporal RFs. This index indicates spatiotemporal asymmetry in space-times RFs and correlates with direction selectivity.^41,54–57^ The mean TDI for the population was 0.33 (0.29 SD), comparable with the ranges in the neural data (mean 0.16; 0.12 SD in cat area 17/18^57^, mean 0.51; 0.30 SD in macaque V1^56^).”

An explanatory paragraph has also been added to Materials and methods section. It reads: “The tilt direction index (TDI)^41,54–57^ of an RF is given by (*R_p_* – *R_q_)/(R_p_* + R_q_), where *R_p_* is the amplitude at the peak of the 2D Fourier transform of the 2D spatiotemporal RF, found at spatial frequency *F*_space_ and temporal frequency *F*_time_. *R_q_* is the amplitude at (*F*_space_, -*F*_time_) in the 2D Fourier transform. The mean and standard deviations of TDI for experimental data for the cat^57^ and macaque^56^ were measured from data extracted from figures in the relevant references (Figure 11A and the low-contrast axis of Figure 3A respectively).”

We also describe the relationship between the spatial and temporal frequency of the model units in the Results section.

12) Figure 7B and Figure 7—figure supplement 2B: The orientation scatter plot is visually difficult to interpret. The quantitative degree of concentration of the preferred orientations on the vertical and horizontal orientations as opposed to the oblique orientations remains unclear. Please provide either an orientation histogram or the percentage of units which fall into the 30-60, 120-150 deg range. In both cases the lowest spatial frequencies should be omitted for the analysis.

A histogram showing the distribution of orientation tuning preferences of the model units has now been added to the Figure 7 and corresponding supplementary figures.

13) Figure 7—figure supplement 1: I am a bit surprised that only 289/400 sparse-coding units can be fitted by a Gabor function. Why is this? Usually most sparse coding units have a good Gabor fit. And with such a high percentage excluded I see the risk of a systematic bias regarding certain tuning parameters.

Sparse coding of images produces mostly Gabor-like filters; however, even in this case, some filters are at the edge of the pixel grid and cannot be reliably fitted. When training on movies, Hyvarinen et al., (2003) also appears to show a similar proportion of units that are not Gabor-like when the full dataset is inspected. In van Hateren and Ruderman, (1998) they only show four example units, so it is difficult to assess how many of the units in the full population are Gabor-like.

14) Figure 7C and Figure 7—figure supplement 2F seem to indicate that the model produces two distinct sub-populations with respect to receptive field parameters n_x and n_y. It should be discussed whether this is the case, and if yes, whether it is systematically related to other parameters (selectivities) of the units. Could this be related related to the two apparent sub-populations regarding separability, blob-like shapes, cf. Figure 4—figure supplement 2? And has such a tendency has also been observed in neural data? In contrast, Ringach, (2004) claimed clustering around a one-dimensional curve. Please describe and discuss in text.

Examination of density plots of these figures suggest that the vast majority of units are spread around a one-dimensional curve, as is the case in Ringach, (2002). However, there is a slight wing formed by a small number of units that extends rightwards from the main curve, but this does not form a distinct separate cluster. We have added a sentence describing this subpopulation to the main text (subsection “Quantitative analysis of visual results”). It reads: “The distribution of units extends along a curve from blob-like RFs, which lie close to the origin in this plot, to stretched RFs with several subfields, which lie further from the origin. A small proportion of the population have RFs with several short subfields, forming a wing from the main curve in Figure 7D.”

As mentioned in response to points 7-9 of the reviewer above, the apparent subpopulation of small blob-like units is no longer present in the new results. Hence, the wing in Figure 7D is not related to this.

15) Figure 8 and subsection “Implementation details: Are we expected to see here that 1600 hidden units are a distinguished optimum? Does this figure not tell us that the prediction error does not substantially depend on the number? And when a biological system could achieve basically the same prediction quality with 100 neurons why should it then invest 1500 additional units for such a small advantage?

As the reviewer points out, in the auditory case, there is not a big change in the performance of the model or in the similarity of its RFs for an increase in the number of units, as can be seen from Figure 8. Nevertheless, a shallow minimum does exist when this parameter is varied and we took the network that gave an absolute minimum as measured by the mean squared error prediction on a held-out validation set so as to be unbiased in our selection process. Furthermore, in the visual case, the number of hidden units seems to play a bigger role both in the network’s performance on the prediction task and on the shapes of the RFs produced. This should now be clear from the additional interactive supplementary figure (Figure 8—Figure Supplement 3; https://yossing.github.io/temporal_prediction_model/figures/interactive_supplementary_figures.html).

16) Figure 8 and Figure 8—figure supplement 1. The comparison between the models prediction capability and the similarity of the model units to real units should not only be presented for the auditory neurons but also for the visual neurons (according to the text the data seem to be already available).

Our lab is primarily focused on auditory processing, hence, we were able to easily obtain recordings from a large population of neurons in A1. However, it was harder to find a large population of spatiotemporal RFs of V1 neurons to compare to, despite requests to multiple labs who focus on visual cortex. We were kindly provided with the spatiotemporal RF data from 8 neurons recorded in V1 of cats by Ohzawa et al., but did not feel that a meaningful quantitative comparison of the kind shown in Figure 8 could be performed with so few neurons.

17) Subsection “Optimising predictive capacity” It is not clear whether the similarity measure (only the span of temporal and frequency tuning is considered) is fair or biased for the comparison with the sparse coding model. What will happen, for example, if the distribution of orientation preference would be used instead (for the visual model)? Would then the sparse coding model appear more similar to the real neurons than the prediction model?Please motivate and discuss.

In comparison to many other papers examining normative models of auditory RFs, we do far more in the way of quantitative comparisons both to other models and to the real data. We took the span of temporal and frequency tuning as reasonable measures of similarity although we concede that other measures could, of course, be chosen. It should be noted that in an attempt to be as fair as possible to the sparse coding model, we did not include measures of similarity (for instance of the proportion of power contained in each time step, as seen in Figure 6B) that seemed to obviously favour our model. Evidence for our hypothesis is also bolstered by the analysis of the effects of multidimensional scaling (Figure 6A), which is a nonparametric measure of similarity.

We have modified the Discussion section to highlight the point raised by the reviewer. The text now reads: “Finally, the more accurate the temporal prediction model is at prediction, the more its RFs tend to be like real neuronal RFs by the measures we use for comparison.”

For the visual data, we did not have sufficient data to compare with the models, particularly in the temporal domain, as we describe above. Quantitative comparison with measurements of data from different datasets and species, recorded using different methods, is no substitute for a single consolidated dataset, which we could not obtain from other labs. Hence, we decided it better to only do this for the auditory dataset and take a more descriptive approach for the visual model.

18) Discussion section: I cannot follow the argumentation that the class of prediction models (or this specific model) should somehow be unique with respect to the ability to provide an independent criterion for the selection of hyperparameters. Should this somehow be a principle, or a logical conclusion? Or an empirical observation only for this special case? Is it logically impossible that we find a measure, for example something entropy-related or whatever, for sparse coding that could have the same status? Please clarify.

It is conceivable that some independent reason for picking the hyperparameters for the sparse coding model could be found that also provides the most neural-like RFs. However, in the previous literature, no such measure is used or proposed. Instead, the hyperparameters, which can greatly affect the nature of the RF structures, are typically chosen in order to produce RFs that look most like the real neural data. Similarly, no obvious measure is apparent to us. To clarify our argument, the following sentence has been added to the end of the paragraph the reviewer refers to (subsection “Optimising predictive capacity”). It reads: “To our knowledge, no such effective, measurable, independent criterion for hyperparameter selection has been proposed for other normative models of RFs.”

19) Subsection “Visual normative models”:does not refer to prediction of the future: Why not? Is not the temporal prediction made in these models that the future spatial pattern is the same as the previous spatial pattern?

Because of the new paragraph in the Introduction that sets out what is meant by predictive coding and temporal prediction, we feel that this paragraph is now largely redundant. The paragraph and sentence to which the reviewer refers has therefore been removed.

20) Subsection “Visual normative models”: The model is selective, throws away information, as opposed to the information preserving properties of other models. But is this really a good strategy for *early* stages of a multi-stage information processing system? Does the principle of least commitment not suggest just the opposite strategy?

It could be true that the structure of sensory RFs is governed by the principle of least commitment, preserving all information. Conversely, it could be the case that temporal prediction is a more important principle governing the structure of sensory RFs. This is an empirical question. We hope that our results provide some good evidence in favour of the latter hypothesis.

21) For the Discussion section it would be of interest to consider the contributions from the two components of the prediction model, i.e., which properties of the units are genuinely caused by prediction and which are more due to the sparse coding part?

We have added a sentence to the Results section clarifying the nature of the sparsity in our model. It reads: “Note that this sparsity constraint differs from that used in sparse coding models, in that it is applied to the weights rather than the activity of the units, being more like a constraint on the wiring between neurons than a constraint on their firing rates.”

We have also examined the specific role of the regularization on the model, as shown in (Figure 8—figure supplement 2 and Figure 8—figure supplement 3;https://yossing.github.io/temporal_prediction_model/figures/interactive_supplementary_figures.html) and discussed this in detail in the responses to the Editor’s second and third points. We find that all three components of the model (prediction, *L*_1_ regularization of the weights and nonlinearity) are essential for providing V1 or A1 like RFs.

[Editors' note: further revisions were requested prior to acceptance, as described below.]

The manuscript has been improved but there are some remaining issues that need to be addressed before acceptance, as outlined below:The authors have addressed my comments in a careful manner. In particular, I appreciate that they made considerable and appropriate changes to text and figures instead of just providing arguments in the response letter. From my view, the manuscript is basically ready for publication. I have a few final suggestions.1).… We examined linear aspects of the tuning of the output units for the visual temporal prediction model using a response-weighted average to white noise input, and found punctate un-oriented RFs that decay into the past..…This is interesting. Can you mention this somewhere in the text?

We have now added the following sentence to the Results section.

“We examined linear aspects of the tuning of the output units for the visual temporal prediction model using a response-weighted average to white noise input, and found punctate non-oriented RFs that decay into the past.”

2) I understand that the model, by its very nature would not care about the sign. But the fact remains that you have an output of a model and you post hoc manipulate this output to obtain a "better suited" presentation (e.g., to ease comparison). My only point is that it should be totally clear to even a superficial reader that such a post hoc change has been applied. So please just include an appropriate sentence that makes this clear, e.g.:Note that the model does not care about the sign (excitation/inhibition) and thus provides no systematic prediction of it. We hence switched the signs of the respective receptive fields of the model output appropriately to obtain receptive fields which all have positive leading excitation.

We have added the following text to the legend of Figure 2D:

“Note that the overall sign of a receptive field learned by the model is arbitrary. Hence, in all figures and analyses we multiplied each model receptive field by -1 where appropriate to obtain receptive fields which all have positive leading excitation (see Materials and methods section).”

(3) Can you mention this alternative goal of least commitment somewhere in the discussion? And the empirical question.

After further consideration our thoughts on this point have become more nuanced. The principle of least commitment requires not doing something that may later have to be undone. Whether the temporal prediction hypothesis is in conflict with least commitment is unclear, and a detailed exploration of this beyond the scope of this paper. According to the temporal prediction hypothesis, aspects of the past which never influence the future will never be of use to an animal, and thus it could be argued that not encoding those aspects will never need to be undone, and hence there is no conflict. However, specific models instantiating the temporal prediction may have limited capacity to identify predictive information, and thus may discard some information that may be useful in the future, and hence may run into conflict with the principle of least commitment. It is also the case that given limited brain capacities, at some point in the brain commitment is required, and the temporal prediction principle may provide a good mechanism to decide what to commit to representing and what to discard. Hence, it is a complicated empirical and theoretical question as to whether and when the principles are in conflict or congruence, and if in conflict under what conditions one is dominant. To reflect this more nuanced view we have how added the following text to the Discussion section:

“There is an open question as to whether the current model may eliminate some information that is useful for reconstruction of the past input or for prediction of higher order statistical properties of the future input, which might bring it into conflict with the principle of least commitment^69^. It is an empirical question how much organisms preserve information that is not predictive of the future, although there are theoretical arguments against such preservation^2^. Such conflict might be remedied, and the model improved, by adding feedback from higher areas or by adding an objective^4–6,60^ to reconstruct the past or present in addition to predicting the future.”